# A Web-Based Decision Support System for Project Evaluation with Sustainable Development Considerations Based on Two Developed Pythagorean Fuzzy Decision Methods

Asad Mahmoudian Azar Sharabiani 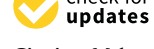 and Seyed Meysam Mousavi *

Department of Industrial Engineering, Shahed University, Tehran 3319118651, Iran;
asad.mahmoudian@shahed.ac.ir
* Correspondence: sm.mousavi@shahed.ac.ir

**Abstract:** Decision support systems are being developed as attractive tools to help organizations make better decisions. These systems assist decision-makers in making the best decisions. The widespread application of the internet has transformed the development of decision support systems into a web-based challenge. On the other hand, project selection has always been a significant issue for organizations. The limitation of resources and the existence of different criteria while selecting projects cause organizations to face the challenges of multiple-criteria decision making. In this research, a new approach is introduced for the selection of criteria. It also presents a new web-based decision support system for selecting projects considering uncertainty and various criteria, including organizational strategies, the seventh edition of project management standard, and sustainable development. Therefore, the economic, social, and environmental dimensions of sustainable development were included as project evaluation indicators. The proposed approach was developed using Pythagorean fuzzy sets, MEREC, and MARCOS methods to examine uncertainty and solution methods. In this approach, a new version of the MARCOS method was developed, with Pythagorean fuzzy sets for rankings. Also, a new development was presented using the Pythagorean fuzzy (PF)-MEREC method, which was used for weighting. The effectiveness of the proposed method is discussed through a real case study conducted on one of the mineral holdings in Iran. Among the mining projects introduced to the company, finally, the second project was selected. In the comparison made using PF-Entropy-TOPSIS and PF-Entropy-VIKOR methods, the superior project provided similar results. By changing the weights of the criteria for four different types of states, sensitivity analysis was used to determine the reliability of the final rankings. In these states, the weights of the criteria were moved together or assigned equal weights, and, in all four states, the ranking results were the same.

**Keywords:** sustainable project management; web-based decision support system; multi-criteria decision making; uncertainty analysis; Pythagorean fuzzy sets; MEREC method; MARCOS method

## 1. Introduction

A project is a temporary task that produces a unique product or service. Temporary tasks have a beginning and an end. After this work is completed, the team either disbands or moves on to new projects. Projects are frequently considered one-shots since they produce distinctive goods or services [1]. Due to the limited nature of resources, selecting the best projects for investment has become a critical issue among decision-makers in organizations, posing a challenge for project-oriented organizations and holdings. The correct selection of projects should lead to the highest possible levels of achievement of organizational goals. In contrast, perceivably, an inaccurate selection of projects could lead to the resources of the organization being spent non-optimally, providing less profit for the organization [2]. The decision-making process is complicated given the large number and complexity of projects, as well as the existence of a large number of qualitative and quantitative criteria [3]. The

prevalence of numerous criteria for selecting projects requires decision-makers to address a multiple-criteria decision-making (MCDM) problem. One of the essential criteria in this research was proper orientation with an organization's strategies. Furthermore, most of the criteria examined in this research were extracted from the seventh edition of the Project Management Body of Knowledge (PMBOK) and the criteria covering sustainable development. Sustainable development is development that meets the needs of the present conditions without jeopardizing the ability of future generations to meet their needs [4].

On the other hand, examining projects according to criteria requires data and expert opinions. Due to the lack of adequate historical data, vagueness, and the large influence of expert judgment on project selection problems, fuzzy set theory has been used as an accepted approach considering project uncertainty [5]. Omeri and Faris [6] developed a mixed b-fuzzy topological space. Classical fuzzy sets cannot reflect the degree of uncertainty. Moreover, they only show the degree of membership and cannot reflect the degree of non-membership. In addition to alleviating the aforementioned shortcomings, Pythagorean fuzzy sets can examine larger spaces than intuitionistic fuzzy sets. Therefore, fuzzy set theory and its extension, Pythagorean fuzzy sets (PFSs), were employed in this research.

In recent years, one of the best approaches for evaluating projects that can process a large number of data and criteria by considering the space of uncertainty is decision support systems. Decision support systems are currently more widely used by decision-makers than before. Given the evident advantages of web-based systems, such as availability for everyone through the internet and not requiring hard drive storage, a new web-based decision support system was developed in this study to help decision-makers select the most productive projects. Furthermore, the authors have sought to employ a novel ranking approach to achieve better results. Many ranking methods have been proposed in the literature over the past few years. The measurement alternatives and ranking according to compromise solution (MARCOS) ranking approach introduced in 2020 and the method based on the removal effects of criteria (MEREC) as a weighting method introduced in 2021 are among the latest techniques that have been employed in the current research regarding Pythagorean fuzzy sets. The advantages of the MARCOS method are its consideration of anti-ideal and ideal solutions at the beginning of the formation of an initial matrix, closer determination of utility degree concerning both solutions, the proposal of a new way to determine utility functions and their aggregation, and the possibility of considering a large set of criteria and alternatives while maintaining the stability of the method [7]. The most important feature of the MEREC method is that a logarithmic function is used to measure the aggregate performance of alternatives; however, as the main advantage of the proposed method, it is flexible so that decision-makers can use different functions for performance calculation [8]. The significant contribution of this study lies in the fact that a new integrative approach was employed to develop a web-based decision support system for evaluating projects by considering Pythagorean fuzzy uncertainty and two new decision-making approaches, i.e., MARCOS and MEREC, while focusing on the principles of PMBOK (7th edition), organization strategies, and sustainable development.

Fallahpour et al. [9] proposed a fuzzy decision support system for a sustainable construction project selection based on FAHP. The selection of project proposals is often modeled as a multiple-criteria decision-making (MCDM) problem. MCDM divides problems into several components. Once decisions are made regarding these parts, they are reassembled to reveal the picture of decision making [10]. If evaluation methods were divided into two numerical and non-numerical categories, non-numerical selection methods would encompass methods such as comparative methods, the most important of which is AHP. Valmohammadi et al. [11] developed a hybrid approach based on FAHP-FTOPSIS for six sigma project selection. The most prevalent numerical selection methods are financial evaluation methods, scoring methods, and mathematical optimization methods, which were the primary scoring methods used in this research. Gulsen et al. [12] assigned a weight value to each parameter using both AHP and fuzzy methods.

Bai et al. [13] proposed a method based on mathematical programming to select a project portfolio and strategy, and Priyalatha et al. [14] and wang et al. [15] used the developed mathematical model in their works. In the other category, MCDM models can be divided into compensatory and non-compensatory groups. Different techniques are used in compensation methods. Pramanik et al. [16] used the integration of fuzzy Shannon entropy and a fuzzy technique (FTOPSIS) for project selection.

MEREC and MARCOS were used as the ranking and weighting methods in this research, respectively. Reviewing the reports of Vakilipour et al. [17], Vassoney et al. [18], Ulubeyli and Kazaz [19], Selmi et al. [20], Alsalem et al. [21], Butkiene et al. [22], Le and Nhieu [23], Mahmoud and Garcia [24], Koç and Gurgun [25], and Pramanik et al. [26] indicated that the MARCOS and MEREC methods had the following advantages over other decision-making methods such as ELECTRE, AHP, SAW, TOPSIS, PROMETHEE, VIKOR, COPRAS, and BWM: (1) allowing for the consideration of a large number of criteria and alternatives [26]; (2) simplicity in calculations and methodological comprehensibility; and (3) high stability [26].

Jalota et al. [27] proposed a decision support system (DSS) for generating a suitable portfolio for an investor in an uncertain multiple-criteria framework and modeled the parameters using L-R fuzzy numbers. Patalay and Bandlamudi [28] proposed a DSS for stock portfolio selection using AI and ML; it was completed using a mathematical optimization model. Xidonas et al. [29] established a decision support system for multiple-criteria portfolio selection using Python. Frej et al. [30] modeled a DSS for project selection under MCDM conditions with mathematical programming. Regarding the literature on organizational strategies, Rowzan [31] considered one of the critical challenges in project-oriented organizations to be the orientation of portfolio management with respect to a company's main strategies.

There have been many studies on the use of fuzzy numbers in decision-making problems. Aghamohagheghi et al. [32] used an interval-valued Pythagorean triangular fuzzy number (IVPTFN) as a tool for solving decision-making problems with ambiguous values, in which the weights of decision-makers were considered. Peng and Ma [33] showed that a Pythagorean fuzzy set (PFS), as a generalization of an intuitionistic fuzzy set (IFS), better captured the uncertainty of experts' decision-making information. Li et al. [34] highlighted that PFSs were among the extensions of intuitionistic fuzzy sets that incorporated more uncertainties to depict fuzzy information; hence, their applications were more comprehensive. Mohagheghi and Mousavi [5] applied Pythagorean fuzzy sets, WASPAS, and MOORA in their research. Komsiyah et al. [35] developed a DSS with ELECTRE and triangular fuzzy sets.

Salimian et al. [36] presented an intuitionistic fuzzy model with an interval-valued approach via VIKOR and MARCOS for selecting a stable supplier in organ transplant networks for health devices. Hashemi et al. [37] applied a decision-making model with multi-criteria analysis conducted by a group of decision-makers (DMs) using intuitionistic fuzzy sets (IFSs) with ELECTERE and VIKOR for a contractor assessment problem. Puska et al. [38] selected sustainable suppliers using triangular fuzzy numbers and the MARCOS method. Kumar et al. [39] developed an integrated BWM fuzzy-MARCOS approach, including BWM and fuzzy-MARCOS, and applied it for the selection of coating materials in tooling industries. Jahangiri [40] analyzed the process of supplying water to the cities and villages of Iran and the disposal of their waste using the MARCOS method. Tas et al. [41] developed a spherical fuzzy SWARA-MARCOS approach for green supplier selection. Mishra et al. [42] explained the purpose of their study as being to integrate Pythagorean fuzzy information-based fairly aggregation operators, criteria importance through intercriteria correlation (CRITIC), pivot pairwise relative criteria importance assessment (PIPRECIA), and MARCOS methods to assess and rank sustainable suppliers in circular supply chains. Ali [43] applied MARCOS and CRITIC methods to the context of q-rung orthopair fuzzy numbers.

Mishra et al. [44] presented an integrated decision support framework using single-valued MEREC-MULTIMOORA for low-carbon tourism strategy evaluators. Simic et al. [45] applied MEREC-COCOSO and FERMAT fuzzy sets in their research. Ghorabaee et al. [8] de-

veloped a new method, called the method based on the removal effects of criteria (MEREC), to calculate the objective weights of attributes. Recently, Zhai et al. [46] assessed the risks posed to the agricultural supply chain for the investment of small and medium-sized agricultural firms using a hybrid model of Pythagorean fuzzy sets and MEREC. Also, in another study, Chaurasiya and Jain [47] introduced a hybrid MCDM method in a Pythagorean fuzzy environment based on a MEREC method (PF-MEREC) and SWARA. Mishra et al. [48] proposed an intuitionistic fuzzy fairly operator and an additive ratio assessment-based integrated model for selecting the optimal sustainable industrial building options. Chaurasiya and Jain [49] introduced an integrated PF-SWARA-MARCOS technique for ranking the selection of the best alternatives in MCDM problems.

The motivation of this research was based on two primary dimensions: sustainability in mining projects and developing a decision-making approach regarding web-based DSS. In this respect, a real case study was performed on the sustainability of the nature of these projects, possessing the elements of a sustainability problem. Furthermore, a new integrated web-based DSS model was developed by combining MEREC and MARCOS decision-making approaches to compute the weights of the criteria and rank the sustainable alternatives in various periods according to internet equipment.

Table 1 shows the differences between this paper and previous research works. As can be seen, the focus of research in recent years has been on the use of MCDM methods with Pythagorean fuzzy sets. Combining the MEREC method with Pythagorean fuzzy sets was performed in 2023; however, these works lacked a decision support system. In other research works mentioned in this table, it can be observed that the integration of MARCOS and MEREC methods with Pythagorean fuzzy sets has been conducted, but the authors did not use web-based decision support systems. Another difference between our work and previous research is that the integration of project evaluation criteria with the standard of project management (PMBOK7) and sustainable development has not been performed. Our work is the only attempt that includes all above considerations. Given the above-mentioned points, in this paper, a new PF-MEREC-MARCOS method is introduced.

**Table 1.** An overview of studies published in recent years.

| Year | Author | Ranking and Weighting Method | Web-Based DSS | Project or Portfolio | Uncertainty | Fuzzy | DSS | Criteria | | | Case Study | DM Weighting |
| | | | | | | | | Organization Strategy | PMBOK 7 | Sustainable Development | | |
|---|---|---|---|---|---|---|---|---|---|---|---|---|
| 2018 | Rowzan [31] | TOPSIS | | * | | | | * | | | | |
| 2019 | Komsiyah et al. [35] | F-ELECTRE | | | * | PFNs | * | | | | | * | |
| 2020 | Fallahpour et al. [9] | FAHP | | * | * | IFS | | | | * | * | |
| 2021 | Valmohammadi et al. [11] | FAHP, FTOPSIS | | * | * | TFN | | | | | | |
| 2021 | Tas et al. [41] | PF-MEREC-SWARA-COPRAS | | | * | PFSs | | | | | * | * |
| 2022 | Salimian et al. [36] | MARCOS-VIKOR | | | * | IVIF | | | | | * | |
| 2022 | Puska et al. [38] | PF-MARCOS | | | * | PFN | | | | | * | |
| 2022 | Chaurasiya and Jain [47] | PF-MEREC-SWARA-COPRAS | | | * | PFSs | | | | | * | * |
| 2023 | Mishra et al. [42] | PF-MARCOS-PIPRECIA-CRITIC | | * | * | PFSs | | | | | * | * |
| 2022 | Mishra et al. [44] | ARAS-MEREC-SWARA | | | * | IFS | | | | * | * | * |
| 2023 | Chaurasia and Jain [49] | PF-SWARA-MARCOS | | | * | PFSs | | | | | * | * |
| **This research** | | PF-MEREC-MARCOS | * | * | * | PFSs | * | * | * | * | * | * |

"*" indicates the realization of the keyword considered in the research.

The main innovations of this paper are represented below:

- Pythagorean fuzzy sets had not, until now, been integrated with MARCOS, MEREC, and web-based DSS approaches in any studies. Although the MEREC-MARCOS method has been combined with Pythagorean fuzzy in recent studies, the score function used in these cases did not consider the degree of uncertainty, a research gap that was addressed in this study.
- Providing a web-based decision support system for implementing the aforementioned methods is an innovative idea that has never been explored before.
- Another innovation of this research is using the principles of the PMBOK, 7th edition, while considering sustainable development to evaluate projects.
- The method proposed in this study involved the performance of computational operations using fuzzy operators until the final steps of the MARCOS method while performing mathematical operations in crisp form in the final steps of this method, an approach that has been shown to increase the accuracy of fuzzy calculations. Also, operators other than the basic operators of Yager [50] have used fuzzy operators of addition, division, and subtraction. Among other cases included in this research, we employed a score function with a degree of doubt.

The research questions addressed in this review are as follows:

Criteria: RQ 1: How has PMBOK7 been used as a project evaluation tool? How is sustainable development related to project evaluation criteria?

MCDM: RQ 2: How have MEREC and MARCOS methods been combined with PFSs and used to evaluate projects?

DSS: RQ 3: How can a decision support system on the web be implemented to evaluate projects?

The remainder of this study is organized as follows: The basic, essential examinations of PFSs are introduced in Section 2. Section 3 proposes a new hybrid soft-computing method based on the combination of MARCOS and MEREC methods under PFS situations, and Section 4 presents the proposed web-based decision support system. In Section 5, a case study on one of the mineral holdings in Iran is discussed. In Section 6, the results of a sensitivity analysis are presented, and Section 7 proposes managerial implications. Finally, Section 8 presents the conclusions and future research suggestions.

## 2. Preliminary Material

In this section, basic definitions and operators of PFS are presented. Uncertainty is a parameter of utmost significance in any decision-making process, especially in multiple-criteria decision making. Uncertainty in decision making has been measured using a variety of methods. Fuzzy logic and its extensions, such as type 2 fuzzy sets, intuitionistic fuzzy sets (IFSs), Pythagorean fuzzy sets (PFSs), neutrosophic sets (NSs), and hesitant fuzzy sets (HFSs), have been developed to address this issue. Pythagorean fuzzy sets are an extension of intuitionistic fuzzy sets and allow decision-makers to use a broader range of membership and non-membership values, with the restriction that the maximum sum of the squares of these values must be 1. Therefore, Pythagorean fuzzy sets provide more flexibility in assigning degrees of membership and non-membership.

Pythagorean fuzzy sets, introduced by Yager in 2013, can be defined with both membership and non-membership degrees and are an excellent solution for addressing uncertainty, where $\mu_{A(x)}$ denotes the degree of membership and $v_{A(x)}$ denotes the degree of non-membership. Their corresponding equations were deemed valid for fuzzy sets; however, before Pythagorean fuzzy sets, intuitionistic fuzzy sets with the restriction condition of $0 \le \mu_{A(x)} + \mu_{A(x)} \le 1$ were introduced. The restriction condition of a Pythagorean fuzzy set is $\mu_{A(x)}^2 + v_{A(x)}^2 \le 1$. Obviously, the second statement results from the first statement. That is, if $\mu_{A(x)} + \mu_{A(x)} \le 1$, then $\mu_{A(x)}^2 + v_{A(x)}^2 \le 1$. Yager showed that Pythagorean membership degree space was larger than the intuitionistic membership degree space. That is, every intuitionistic fuzzy set is a Pythagorean fuzzy set [51]. Figure 1 compares the intuitionistic and Pythagorean fuzzy spaces. The space bounded by the curved line

refs to Pythagorean fuzzy space, while the space bounded by the straight line represents intuitionistic fuzzy space, which is clearly smaller.

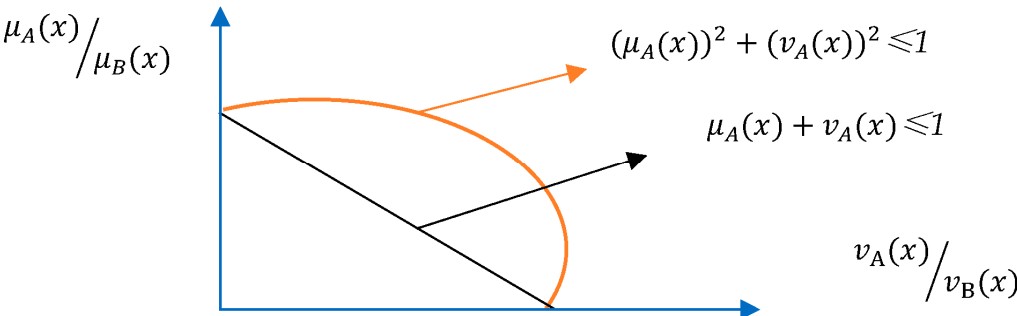

**Figure 1.** Intuitionistic and Pythagorean fuzzy spaces [50].

**Definition 1.** *For each x, the degree of uncertainty is defined as $\pi_{p(x)}$, which reflects a lack of knowledge about whether or not x belongs to set A. This degree of doubt in the intuitionistic fuzzy set can be formulated as follows:*

$$\pi_{p}(X) = \mu_A(X) - v_A(X) \tag{1}$$

For Pythagorean fuzzy sets, the degree of uncertainty can be formulated as follows:

$$\pi_{p}(X) = \sqrt{1 - \left[ (\mu_A(X))^2 + (v_A(X))^2 \right]} \tag{2}$$

Lower values of $\pi_{p(x)}$ indicate higher certainty regarding $x$, and vice versa [52]. Regarding the basic operators of Pythagorean fuzzy numbers, according to the results reported by Yager [51], the following equations could be formulated.

**Definition 2.** *If $\beta_1 = P_{(\mu_{\beta1}, v_{\beta1})}$ and $\beta_2 = P_{(\mu_{\beta2}, v_{\beta2})}$ are two Pythagorean fuzzy sets, $\lambda > 0$ is a scalar number. The basic operators can be defined via the following equations:*

$$\beta1 \oplus \beta2 = P\left( \mu_{\beta1}^2 + \mu_{\beta2}^2 - \mu_{\beta1}^2\mu_{\beta2}^2, \; v_{\beta1} v_{\beta2} \right) \tag{3}$$

$$\beta1 \otimes \beta2 = P(\mu_{\beta1}\mu_{\beta2}, \; \sqrt{v_{\beta1}^2 + v_{\beta2}^2 - v_{\beta1}^2 v_{\beta2}^2}) \tag{4}$$

$$\lambda\,\beta = P\left( \sqrt{1 - (1 - (\mu_\beta^2))^\lambda}, v_\beta^\lambda \right) \tag{5}$$

$$\beta^\lambda = p\left( \mu_\beta^\lambda, \sqrt{1 - (1 - (v_\beta^2))^\lambda} \right) \tag{6}$$

**Definition 3.** *Owing to the limitation of primary division and subtraction operators, newer division and subtraction operators introduced by Peng et al. [53] were used here, which are presented in Equations (7) and (8).*

$$\beta1 \oslash \beta2 = \left( \sqrt{\mu_{\beta1}^2 + v_{\beta2}^2 - \mu_{\beta1}^2 v_{\beta2}^2}, \; v_{\beta1} \; \mu_{\beta2} \right) \tag{7}$$

$$\beta1 \ominus \beta2 = \left( \mu_{\beta1} \; v_{\beta2}, \; \sqrt{v_{\beta1}^2 + \mu_{\beta2}^2 - v_{\beta1}^2\mu_{\beta2}^2} \right) \tag{8}$$

**Definition 4.** *The score function from Peng et al. [53], which accounted for the degree of uncertainty, was used in the current study according to Equation (9). The most important advantage of this score function is that, in addition to μ and v, it also uses the degree of uncertainty.*

$$s_p = \frac{e^{\mu^2 - v^2}}{\pi^2 + 1} \tag{9}$$

**Definition 5.** *To aggregate PFNs, Yager [51] introduced a Pythagorean fuzzy weighted averaging (PFWA) aggregation operator, which does not conform to the basic operating rules of the PFNs presented in Yager's work [51]. Although the aforementioned operator was simple and easy to use for aggregating PFNs, a new PFWA operator was introduced based on the operational rules of PFNs defined by Zhang [54], and it was used in the current work.*

$$\text{PFWA } (\beta_1, \ \beta_2, \ \ldots, \ \beta_n) = w_1\beta_1 \oplus w_2\beta_2 \oplus \cdots \oplus w_n\beta_n = \text{p}$$
$$(\sqrt{1 - \prod_{j=1}^{n}(1 - (\mu_{\beta_j})^2)^{\overline{w_j}}}, \ \prod_{j=1}^{n}\left(v_{\beta_j}\right)^{w_j}) \tag{10}$$

*Here, $w_j$ denotes the importance degree of $\beta_j$ satisfying $w_j \geq 0$ (j = 1, 2, ..., n) and $\sum_{j=1}^{n} w_j = 1$.*

### 3. Proposed Soft-Computing Model

The approach proposed in this study consists of five general steps. Figure 2 shows the proposed general approach. The criteria and experts were identified in the first step of this approach. This step concerned the strategies of the organization, the guidelines of PMBOK (7th edition), and sustainable development. The second step pertained to assigning weights to experts and criteria, during which a MEREC method was used to weigh the criteria. The MARCOS method was applied in the third step for scoring purposes, including constructing a decision matrix, constructing a Pythagorean fuzzy weighted matrix, calculating ideal and anti-ideal matrices, constructing normalized and weighted matrices, determining the desirability of alternatives, and determining the final rank. Integration with Pythagorean fuzzy sets was implemented in all steps of this research. In the fourth step, a web-based decision support system was developed using the outputs of the previous step. The final step involved the examination of a case study followed by validation and sensitivity analysis.

The proposed approach is depicted in the flowchart in Figure 3. This flowchart illustrates the process of execution. First, the criteria were identified. The criteria used to evaluate projects should be carefully determined. In this research, the criteria were extracted from PMBOK version 7 and the dimensions of sustainable development. In the following step, weightings were assigned. This operation was carried out with respect to two areas: weighting decision makers and weighting criteria using MEREC method. Then, ranking was conducted using the MARCOS method. The next step was to implement the developed methodology by preparing a decision support system on the web, and the final step was to implement it in the form of a case study and sensitivity analysis.

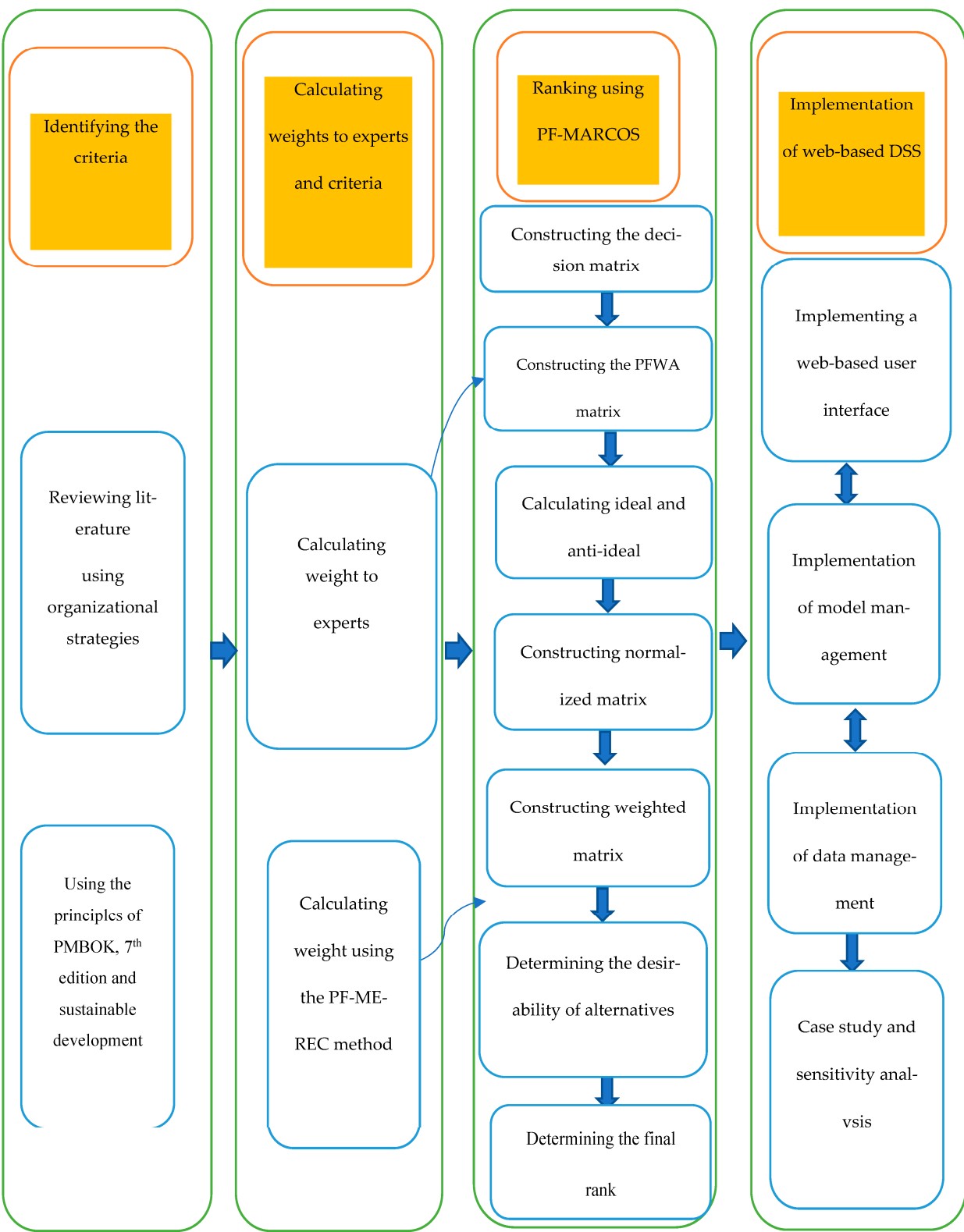

**Figure 2.** The proposed approach.

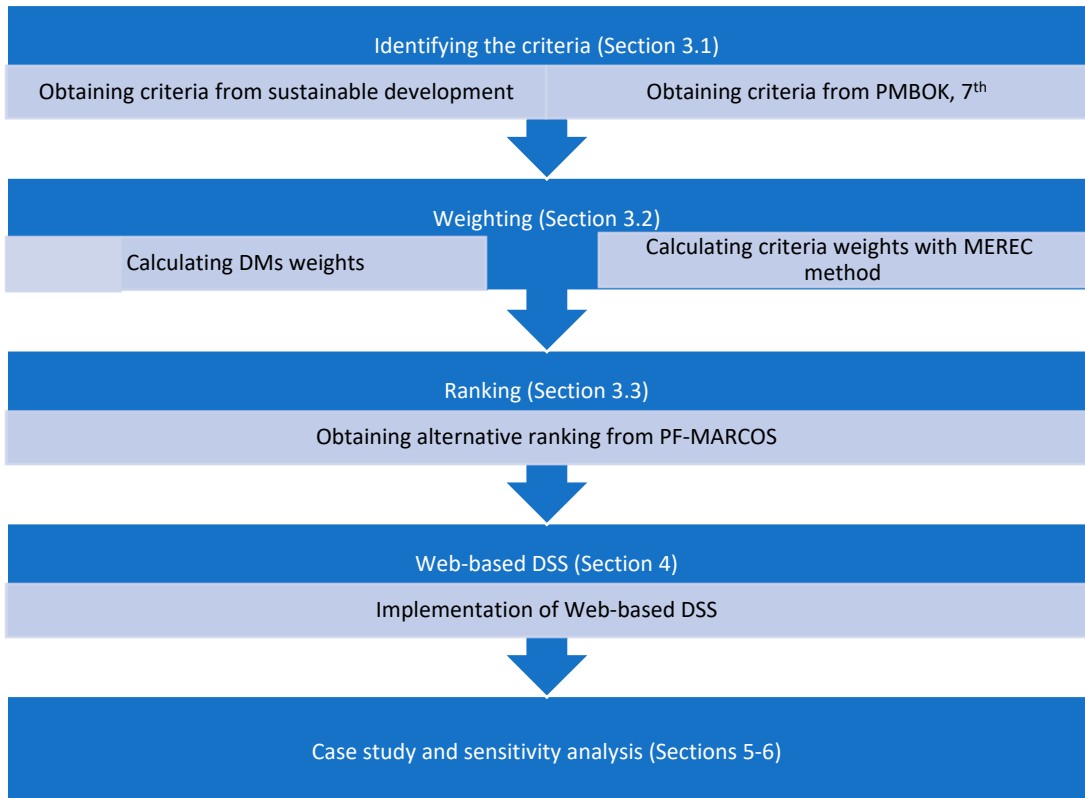

**Figure 3.** Flowchart of research.

### 3.1. Determining the Criteria

Economic analysis is the most commonly applied criteria of investment assessment in classic decision-making procedures [55]. The determination of indicators for evaluating sustainable development in countries was first introduced at the United Nations conference in Rio de Janeiro in 1992 under the title of Agenda 21, and the three main dimensions of sustainable development were identified to be economic, social, and environmental [56]. Sustainability, which proposes the balance of the economic, social, and environmental effects of an investment, is a modern approach.

The sixth edition of PMBOK discusses a process-oriented perspective for project management approaches, while in the seventh edition of PMBOK, this approach is replaced with 12 principles for a project [57]. The authors of the current study have tried their best to consider some of the principles of PMBOK and three general dimensions of sustainability as criteria for collecting the desired criteria with which to review projects.

### 3.2. Calculating Weights of Experts and Criteria

The weights of decision-makers were calculated in this step. The approach proposed in this research entailed group decision-making, for which the method developed by Chaurasiya and Jain [47] was used:

$$e_k = \frac{\mu^2{}_{A(x)} + \pi^2{}_{p(x)}\left(\dfrac{\mu^2{}_{A(x)}}{v^2{}_{A(x)} + \mu^2{}_{A(x)}}\right)}{\sum_{k=1}^{l}\left(\mu^2{}_{A(x)} + \pi^2{}_{p(x)}\left(\dfrac{\mu^2{}_{A(x)}}{v^2{}_{A(x)} + \mu^2{}_{A(x)}}\right)\right)} \tag{11}$$

where $\sum_{k=1}^{l} e_k = 1$.

Weights of the criteria were calculated using the MEREC method. The MEREC method, first introduced in 2021, was used to weight the criteria, the steps of which are presented as follows:

Step 1. Constructing the decision matrix and calculating the score of each option for each criterion.

$$
d_E = \left( d_{ij}^E \right)_{m*n} = \begin{pmatrix} d_{11}^E & \cdots & d_{1n}^E \\ \vdots & \ddots & \vdots \\ d_{m1}^E & \cdots & d_{mn}^E \end{pmatrix}_{m*n}
$$
$$
= \begin{pmatrix} \left( \mu_{d_{11}}^E, v_{d_{11}}^E \right) & \cdots & \left( \mu_{d_{1n}}^E, v_{d_{1n}}^E \right) \\ \vdots & \ddots & \vdots \\ \left( \mu_{d_{m1}}^E, v_{d_{m1}}^E \right) & \cdots & \left( \mu_{d_{mn}}^E, v_{d_{mn}}^E \right) \end{pmatrix}_{m*n}
\tag{12}
$$

A PFWA matrix was constructed. For this purpose, Equation (10) was employed. In this equation, $w_j$ is the weight of decision-makers.

Step 2. Normalizing the decision matrix. The elements of the normalized matrix are given by Equation (13) [45].

$$
N_E = \left( N_{ij}^E \right)_{m*n} = \begin{pmatrix} N_{11}^E & \cdots & N_{1n}^E \\ \vdots & \ddots & \vdots \\ N_{m1}^E & \cdots & N_{mn}^E \end{pmatrix}_{m*n} =
$$

$$
\begin{matrix} C1 & \dots & Cn \end{matrix}
$$
$$
\begin{pmatrix} \left( \mu_{d_{11}}^E, v_{d_{11}}^E \right) & \cdots & \left( \mu_{d_{1n}}^E, v_{d_{1n}}^E \right) \\ \vdots & \ddots & \vdots \\ \left( \mu_{d_{m1}}^E, v_{d_{m1}}^E \right) & \cdots & \left( \mu_{d_{mn}}^E, v_{d_{mn}}^E \right) \end{pmatrix}_{m*n}
\tag{13}
$$

$$
N_E = \begin{cases} \left( v_{d_{ij}}^E, \mu_{d_{ij}}^E \right) & if\ j \text{€} B, Benefit\ creteria \\ \left( \mu_{d_{ij}}^E, v_{d_{ij}}^E \right) & if\ j \text{€} C, Cost\ criteria \end{cases}
$$

Instead of using conventional methods, normalization was conducted using the MEREC normalization method. Switching between equations for advantageous and non-beneficial criteria made differences. All of the criteria were transformed into minimization-type criteria, differentiating this approach from other research works.

Step 3. Calculating the overall performance of alternatives using Equation (14). To determine the overall performance of the alternatives, a logarithmic metric with equal weights for the criterion was used in this step. According to the normalized values obtained from the previous step, one could ensure that smaller values of $n_{ij}^x$ yielded greater performance values ($S_i$). The value of $m$ represents the number of criteria. The following relation was devised for this computation:

$$
S_i = ln(1 + (\frac{1}{m} \sum_j \left| ln \left( n_{ij}^x \right) \right|))
\tag{14}
$$

Step 4. Calculating the performance of alternatives by removing the effects of criteria using Equation (15). This step differed from step 3 in that the alternatives' performances were determined by deleting each criterion individually. The overall performance of the *i*th alternative concerning the removal of the *j*th criterion was denoted as $S_{ij}^/$. Therefore, we had *m* sets of performances associated with *m* criteria.

$$S'_{ij} = ln(1 + (\frac{1}{m} \sum_{k, j \neq k} \left| ln\left(n^x_{ij}\right)\right|)) \tag{15}$$

Step 5. Computing the total absolute deviations using Equation (16). In this step, we calculated the removal effect of the *j*th criterion depending on the results of steps 3 and 4.

$$E_J = \sum_J \left| S'_{ij} - S_i \right| \tag{16}$$

Step 6. Determining the final weights of the criteria using Equation (17). The elimination effects ($E_j$) of step 5 were used to calculate the objective weight of each criterion. In the equation, $w_j$ stands for the weight of the *j*th criterion.

$$w_j = \frac{E_j}{\sum_k E_k} \tag{17}$$

*3.3. Ranking Using the PF-MARCOS Method*

The MARCOS method is among the relatively novel methods of multiple criteria decision making, where alternatives are evaluated and ranked based on a compromise solution. This method is presented as follows:

Step 1. Formulating the decision matrix using Equation (12).

Step 2. Determining ideal and anti-ideal solutions using Equation (18). The anti-ideal solution ($AA_I$) was the worst alternative, while the ideal solution ($A_I$) was a substitute with the most advantageous quality. *B* represents a group of benefit criteria, while *C* represents a group of cost criteria.

$$A_I = MAX_i \ x_{ij} \ if \ j\epsilon B \ , \ MIN_i x_{ij} \ if \ j\epsilon$$
$$AA_I = MIN_i \ x_{ij} \ if \ j\epsilon B \ , \ MAX_i x_{ij} \ if \ j\epsilon C \tag{18}$$

Step 3. Normalization was conducted in this step. The normalized matrix's components were determined using Equation (19).

$$N_E = \left(N^E_{ij}\right)_{m*n} = \begin{pmatrix} N^E_{11} & \cdots & N^E_{1n} \\ \vdots & \ddots & \vdots \\ N^E_{m1} & \cdots & N^E_{mn} \end{pmatrix}_{m*n} =$$

$$\begin{array}{ccc} C1 & \ldots & Cn \end{array}$$
$$\begin{pmatrix} \left(\mu^E_{d_{11}}, v^E_{d_{11}}\right) & \cdots & \left(\mu^E_{d_{1n}}, v^E_{d_{1n}}\right) \\ \vdots & \ddots & \vdots \\ \left(\mu^E_{d_{m1}}, v^E_{d_{m1}}\right) & \cdots & \left(\mu^E_{d_{mn}}, v^E_{d_{mn}}\right) \end{pmatrix}_{m*n}$$

$$N_E = \begin{cases} \left(v^E_{d_{ij}}, \mu^E_{d_{ij}}\right) & if \ j\epsilon C, \ Cost \ criteria \\ \left(\mu^E_{d_{ij}}, v^E_{d_{ij}}\right) & if \ j\epsilon B, Benefit \ creteria \end{cases} \tag{19}$$

Step 4. Using the weight obtained from the MEREC method and forming a weighted matrix using Equation (20).

$$V_{ij} = w_j \times N_E \ : \ P\left(\sqrt{1 - (1 - (\mu^2_\beta))^{w_j}}, v^{w_j}_\beta\right) \tag{20}$$

Step 5. Calculating the utility degree of alternatives $K_i$. Using Equations (21) and (22), the utility degrees of an alternative concerning the anti-ideal ($k^-_i$) and ideal ($k^+_i$) solutions were

determined. In the following equations, $s_i$, ($i = 1, 2, 3, \ldots, m$) is the sum of the values of each row in the weighted matrix, which can be obtained from the following equations:

$$k_i^+ = \frac{s_i}{s_{AI}} : s_i \oslash s_{AI} = \left( \sqrt{\mu_{s_i}^2 + v_{s_{AI}}^2 - \mu_{s_i}^2 v_{s_{AI}}^2} , \ v_{s_i} \ \mu_{s_{AI}} \right) \tag{21}$$

$$k_i^- = \frac{s_i}{s_{AAI}} : s_i \oslash s_{AAI} = \left( \sqrt{\mu_{s_i}^2 + v_{s_{AAI}}^2 - \mu_{s_i}^2 v_{s_{AAI}}^2} , \ v_{s_i} \ \mu_{s_{AAI}} \right) \tag{22}$$

$$S_i = \sum_{j=1}^{n} v_{ij} , \ (i = 1, \ldots, m) : \left( \mu_{\beta 1}^2 + \mu_{\beta 2}^2 - \mu_{\beta 1}^2 \mu_{\beta 2}^2 , \ v_{\beta 1} \ v_{\beta 2} \right) \tag{23}$$

Step 6. Determining the utility function of alternatives $f_{(k)_i}$. The utility function is the compromise of the observed alternative concerning the ideal and anti-ideal solutions determined using Equation (24)

$$f_{(k)_i} = \frac{k_i^+ + k_i^-}{1 + \dfrac{1 - f_{k_i}^+}{f_{k_i}^+} + \dfrac{1 - f_{k_i}^-}{f_{k_i}^-}} \tag{24}$$

where

$$f_{k_i^-} = \frac{k_i^+}{k_i^+ + k_i^-} \tag{25}$$

$$f_{k_i^+} = \frac{k_i^-}{k_i^+ + k_i^-} \tag{26}$$

where $f_{k_i^-}$ represents the utility function concerning the anti-ideal solution, while $f_{k_i^+}$ represents the utility function concerning the ideal solution. Equations (25) and (26) were used to calculate utility functions concerning ideal and anti-ideal solutions.

Step 7. Ranking the alternatives. The ultimate values of the utility functions were used to rank alternatives. It was preferable for an alternative to have the highest feasible utility function value.

One of the properties of the proposed approach was that it employed fuzzy operators in Equations (20)–(23) and Equations (25) to (26) in the DSS model database, and only Equation (24) employed a score function with definite numbers. That is, the current research made use of crisp numbers at the end of the work. This maintained the fuzzy calculations' accuracy until the MARCOS method's final steps.

## 4. Proposed DSS

Decision support systems consist of main components, called a user interface, a database, and a model base. Figure 4 illustrates the proposed DSS framework and the main components of the framework. It shows the models that managers can resort to while decision making. The database contains information from internal and external sources, and the user interface encompasses instruments that help the end user of a DSS navigate through the system. The user interacts with the system through the user interface and thereby inputs information such as the criteria collected from various sources, PMBOK data, language variables, decision makers' specifications, and project information into the system. All data are recorded and stored in the system database. The data inputted into the system through the user interface then become valuable information following their integration into the database and the subsequent processing of the model database. This information includes the weights of decision makers, the weights of criteria, and the final ranking of projects. Finally, the user interface can receive the final reports. An exchange is conducted between the user and the system. The system delivers the outputs after receiving the inputs from the user.

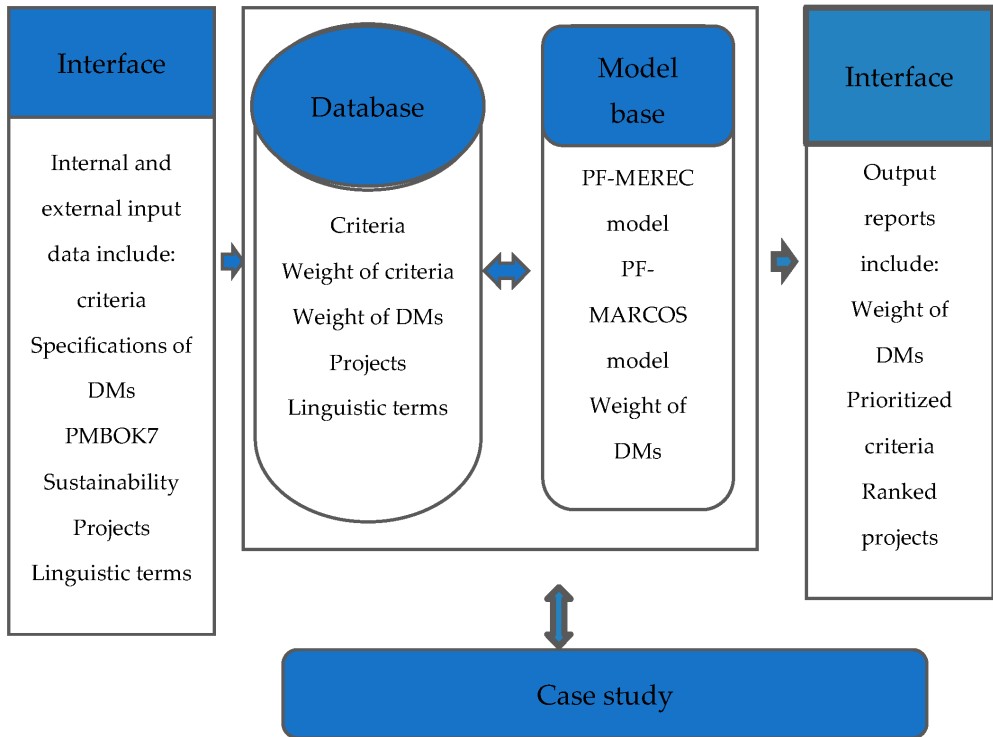

**Figure 4.** The proposed DSS framework.

The use of software systems among users on the web is expanding. Today, the internet has significantly changed the way in which software is used. Software can be used in various forms, and organizations employ multiple models for implementing applications tailored to their needs. Due to the ever-increasing scale of internet use, web-based applications have become widely popular. Any system that can be used on the internet is called web-based software. The other properties and technical specifications of web-based applications include that they are always available for users and customers via different internet browsers such as Chrome and Firefox. That is, customers do not need to have the application installed on their hard drives. Furthermore, these applications use a stable central core, which is available for all users to apply any changes thereon. The primary and most crucial difference between web-based and Windows-based decision support systems is that the latter are installed on a central system, but the former are only executed on web servers and accessible over the internet. Ultimately, Windows applications can only be used on devices using Windows as an operating system and hence cannot be accessed on devices using Mac, Android, Linux, and other operating systems.

Since DSSs are computer-based applications designed to help decision-makers and given that the DSS used in this research is web-based, Figure 5 shows the proposed web-based DSS. Users should search for the DSS address in the web environment of a computer or device connected to the internet, and their requested information will be sent to the DSS server. The server provides a web-based user interface written in C# using the Microsoft ASP.NET framework. This allows people to easily access the server from remote locations. The data related to DSS in this research are based on a database using SQL Server 2019, which was installed on the same DSS server. The requested information was called from the database and provided to the end user. The database type was relational. This relationship between the main tables is shown in Figure 6. In a relational database, relationships are created between tables to avoid repeating multiple records or fields. The main tables of the relational database used herein were Projects, Criteria, DecisionMaking, DecisionMakers, Linguistic_Terms, and DM_Weight. Each table contained information fields, as shown in the figure below.

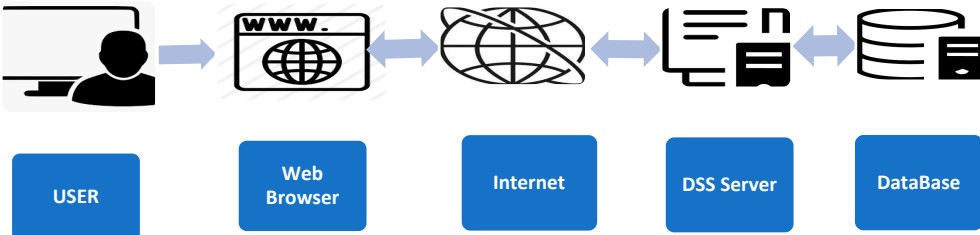

**Figure 5.** Web-based DSS.

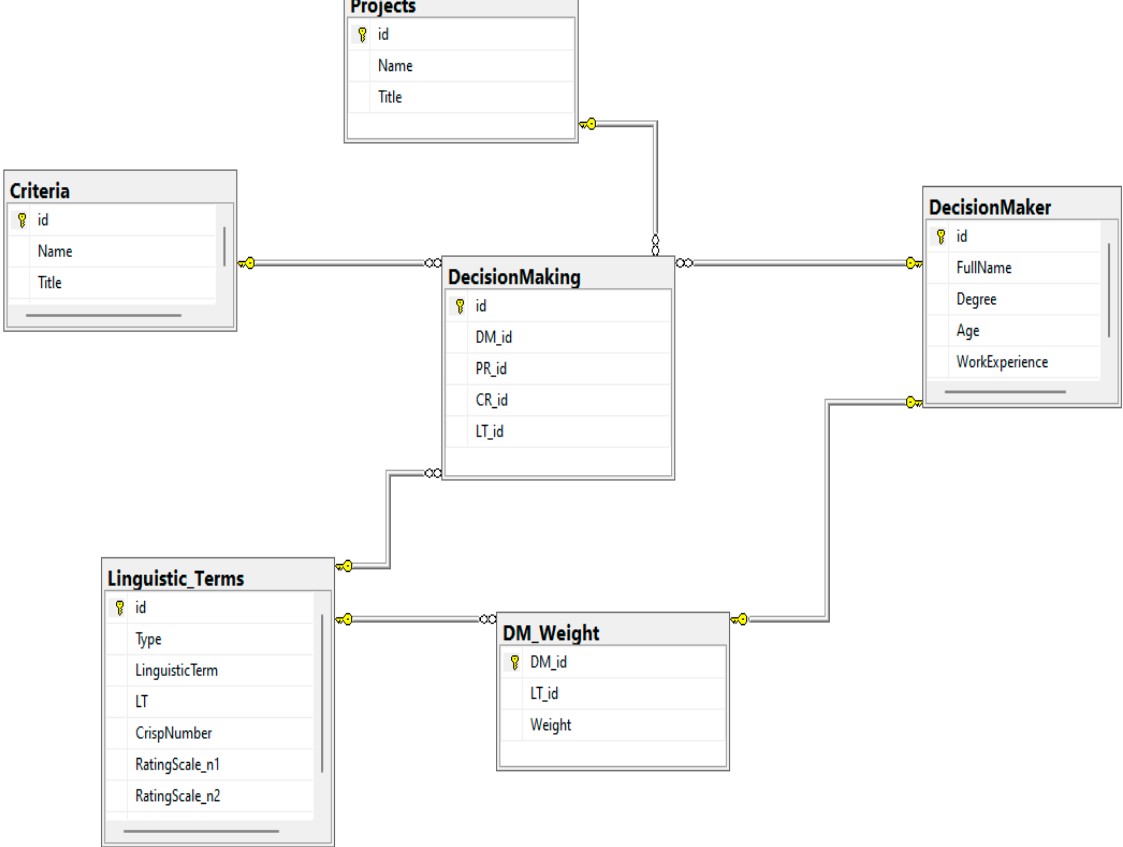

**Figure 6.** Relationships between the main tables in DSS.

To call commands faster in DSS, functions were used. One of the most essential applications of the functions in this DSS was their use for dealing with basic operators. The functions related to basic operators are shown in Appendix A.

## 5. Case Study for Mining Projects

A real case study was performed to examine the efficacy of the proposed approach. This study was carried out to select mining projects for introduction in a specialized holding's portfolio. Although our proposed methodology can be used for all types of projects, because the case study was an actual study, a specialized holding active in the field of mines in Iran was chosen for this purpose. While the applicability process of the mining projects of this holding was focused on the financial criteria of the candidate projects, our proposed solution, with greater comprehensiveness, was welcomed by the decision-makers. The proposed methodology for choosing the best project was developed with the help of this holding. Before, these holding projects were mainly selected by considering a small number of indicators, and the most important indicators were the financial statuses of the candidate projects. The proposed methodology was applied to evaluate the projects by

considering more comprehensive and different criteria. At this point, choosing one of the three projects introduced to a holding was the main issue for decision-makers. These three mines are shown in Figure 7. All three mines were open pit mines and needed investors for development.

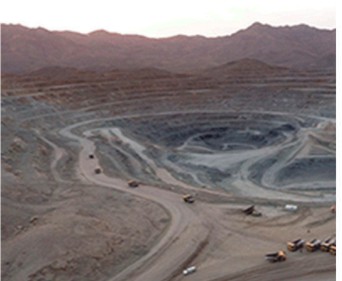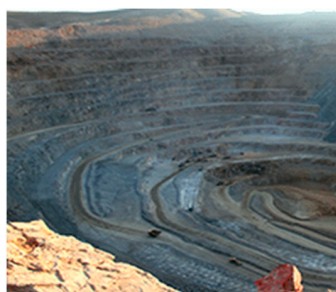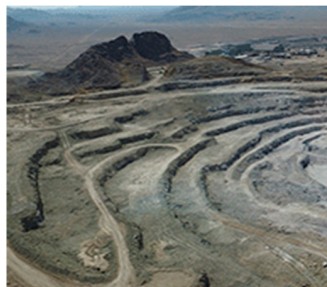

**Figure 7.** Candidate mines.

The first project was a development and exploitation project of an abandoned phosphate mine; this project is denoted as P1. The second project was related to a lead and zinc mine; the proposal for the development of this mine was presented to the holding, and this project is denoted as P2. The third project was a development project of iron ore mining; it is denoted as P3. In this period, a holding could use one of the mines and wanted to choose the mine that would bring the greatest benefit to the holding.

The SQL SERVER 2019 database was used for the DSS designed for data management, while C# was used to manage the user interface. Furthermore, PF-MEREC-MARCOS was used to manage the model in DSS. The DSS could be accessed by professionals at WWW.WDSS.IR.

For this purpose, three experts commented on these projects. The corresponding literature, and, in particular, PMBOK 7th edition, were examined and analyzed to extract the following criteria for project evaluation. Considering the strategies of the organization and general dimensions of sustainable development, the following criteria were used in this research:

1. Strategies; 2. stakeholders; 3. value; 4. environmental sustainability; 5. changes; 6. risk; 7. complexity; 8. customization; 9. economic sustainability; 10. social sustainability.

The presented DSS consisted of four main parts. These sections constituted Admin Area, Basic Data, Data Entry, and Report.

*Admin area:* In this section, different user accesses were set in the form of different roles. Figure 8 shows a view of role page access. This figure shows which parts each user will have access to and the extent of this access. Users were managed in this section. Some issues, like adding a user and determining their access level, as well as password changes, were addressed in this section. User access could be managed in the role page access section.

*Basic data:* Basic information was managed in this section of the DSS. This part included sections such as projects, decision-makers, decision-makers' weights, decision matrix linguistic terms, decision-makers' linguistic terms, and criteria. The information and weights of the decision-makers are shown in Figures 9 and 10. Basic information, such as project information, was entered in the project section. The decision-makers' information was entered in the decision-makers section. The information on language variables was entered in two parts. Linguistic variables related to decision-makers were stored in the decision matrix linguistic terms section, and information related to the linguistic variables of decision-makers' weights was stored in the decision-makers' linguistic terms section. Branch information was entered in the criteria section. Decision-makers were weighted in the decision-makers' weights section.

| New | Role | Page Title | View | Add | Edit | Delete |
|---|---|---|---|---|---|---|
| Edit Delete | Admin | AdminArea | ✔ | ✔ | ✔ | ✔ |
| Edit Delete | Admin | BasicData | ✔ | ✔ | ✔ | ✔ |
| Edit Delete | Admin | DataEntry | ✔ | ✔ | ✔ | ✔ |
| Edit Delete | Admin | Reports | ✔ | ✔ | ✔ | ✔ |
| Edit Delete | Decision Maker | BasicData | ✔ | ✔ | ✔ | ✔ |
| Edit Delete | Decision Maker | DataEntry | ✔ | ✔ | ✔ | ✔ |
| Edit Delete | Decision Maker | Reports | ✔ | ✔ | ✔ | ✔ |
| Edit Delete | Viewer | BasicData | ✔ | ☐ | ☐ | ☐ |
| Edit Delete | Viewer | DataEntry | ✔ | ☐ | ☐ | ☐ |
| Edit Delete | Viewer | Reports | ✔ | ☐ | ☐ | ☐ |

**Figure 8.** Role page access.

| New | Full Name | Degree | Age | WorkExperience | JobPosition |
|---|---|---|---|---|---|
| Edit Delete | MAHMOUD.B | MA | 52 | 28 | |
| Edit Delete | ALIREZA.KH | Masters | 45 | 23 | |
| Edit Delete | HAMID.M | PHD | 39 | 19 | |

**Figure 9.** Decision makers' information.

| New | Decision Maker | Linguistic Term | Weight |
|---|---|---|---|
| Edit Delete | MAHMOUD.B | Very High | 0.341526825809501 |
| Edit Delete | ALIREZA.KH | High | 0.316946348380997 |
| Edit Delete | HAMID.M | Very High | 0.341526825809501 |

**Figure 10.** Decision makers' weights.

Figure 11 shows the data entry environment for criteria. As is apparent in the figure, risk and complexity criteria were regarded as negative criteria, and the remaining criteria were positive. The ability to add criteria and determine whether they were positive or negative is considered in this table.

| New | Criteria Name | Criteria Title | Positive |
|---|---|---|---|
| Edit Delete | ST | Stakeholders | ✔ |
| Edit Delete | VA | Value | ✔ |
| Edit Delete | EN | Environment | ✔ |
| Edit Delete | TA | Tailoring | ✔ |
| Edit Delete | CH | Change | ✔ |
| Edit Delete | RI | Risk | ☐ |
| Edit Delete | CO | Complexity | ☐ |
| Edit Delete | STR | Strategy | ✔ |
| Edit Delete | SO | Social | ✔ |
| Edit Delete | EC | Economic | ✔ |

**Figure 11.** Data for criteria in DSS.

*Data entry*: In this section, decision matrix information was completed. A view of the data entry section is shown in Figure 12. This table shows how to complete the decision matrix. For each project and criterion, linguistic terms were assigned.

*Report:* This section included two parts: report and sensitivity analysis. In the report section, reports of various methods and calculations could be used, and in the sensitivity analysis section, analyses of the indicators' weight could be performed.

In this research, linguistic terms were used based on Tables 2 and 3. Table 2 was used for decision matrices, and Table 3 was used for weighting the experts. The linguistic terms for evaluating decision-makers were chosen in five intervals, and the linguistic terms for forming the decision matrix were selected in seven intervals.

**Table 2.** Pythagorean fuzzy linguistic terms for decision matrix [58].

| Linguistic Term | LT | Crisp Number | Rating Scale ($\mu,v$) | $\pi$ |
|---|---|---|---|---|
| Extremely Low | EL | 0 | (0.15, 0.85) | 0.5 |
| Very Low | VL | 1 | (0.25, 0.75) | 0.61 |
| Low | L | 2 | (0.35, 0.65) | 0.68 |
| Fair | M | 4 | (0.55, 0.45) | 0.7 |
| High | H | 6 | (0.65, 0.35) | 0.68 |
| Very High | VH | 7 | (0.75, 0.25) | 0.61 |
| Extremely High | EH | 8 | (0.85, 0.15) | 0.51 |

| New | Project Title | Criteria Title | Linguistic Term | Rating Scale |
|---|---|---|---|---|
| Edit | P1 | Stakeholders | Fair | ( 0.55 , 0.45 ) |
| Edit | P1 | Value | Fair | ( 0.55 , 0.45 ) |
| Edit | P1 | Environment | Very Low | ( 0.25 , 0.75 ) |
| Edit | P1 | Tailoring | High | ( 0.65 , 0.35 ) |
| Edit | P1 | Change | High | ( 0.65 , 0.35 ) |
| Edit | P1 | Risk | Very High | ( 0.75 , 0.25 ) |
| Edit | P1 | Complexity | Very High | ( 0.75 , 0.25 ) |
| Edit | P1 | Strategy | High | ( 0.65 , 0.35 ) |
| Edit | P1 | Social | Fair | ( 0.55 , 0.45 ) |
| Edit | P1 | Economic | Fair | ( 0.55 , 0.45 ) |
| Edit | P2 | Stakeholders | Fair | ( 0.55 , 0.45 ) |
| Edit | P2 | Value | Fair | ( 0.55 , 0.45 ) |
| Edit | P2 | Environment | Fair | ( 0.55 , 0.45 ) |
| Edit | P2 | Tailoring | Fair | ( 0.55 , 0.45 ) |
| Edit | P2 | Change | Fair | ( 0.55 , 0.45 ) |
| Edit | P2 | Risk | Low | ( 0.35 , 0.65 ) |
| Edit | P2 | Complexity | Fair | ( 0.55 , 0.45 ) |
| Edit | P2 | Strategy | Fair | ( 0.55 , 0.45 ) |
| Edit | P2 | Social | Fair | ( 0.55 , 0.45 ) |
| Edit | P2 | Economic | Extremenly Hight | ( 0.85 , 0.15 ) |
| Edit | P3 | Stakeholders | Very Low | ( 0.25 , 0.75 ) |
| Edit | P3 | Value | Fair | ( 0.55 , 0.45 ) |
| Edit | P3 | Environment | Fair | ( 0.55 , 0.45 ) |
| Edit | P3 | Tailoring | Very Low | ( 0.25 , 0.75 ) |
| Edit | P3 | Change | Fair | ( 0.55 , 0.45 ) |
| Edit | P3 | Risk | Fair | ( 0.55 , 0.45 ) |
| Edit | P3 | Complexity | High | ( 0.65 , 0.35 ) |
| Edit | P3 | Strategy | Fair | ( 0.55 , 0.45 ) |
| Edit | P3 | Social | Fair | ( 0.55 , 0.45 ) |
| Edit | P3 | Economic | Fair | ( 0.55 , 0.45 ) |

**Figure 12.** Data entry in DSS for decision matrix.

**Table 3.** Pythagorean fuzzy linguistic terms for weighting experts [58].

| Linguistic Term | LT | Crisp Number | Rating Scale ($\mu$,$v$) | $\pi$ |
|---|---|---|---|---|
| Very Low | VU | 0 | (0.15, 0.85) | 0.51 |
| Low | U | 1 | (0.35, 0.65) | 0.68 |
| Fair | M | 2 | (0.55, 0.45) | 0.7 |
| High | I | 3 | (0.75, 0.25) | 0.61 |
| Very High | VI | 4 | (0.85, 0.15) | 0.51 |

The initial matrix of the experts' opinions is shown in Table 4. In this table, the opinions of three decision-makers regarding three projects are given. The layout of this table is based on the linguistic terms.

**Table 4.** Initial decision matrix.

| DM | Projects Title | Stakeholders | Value | Environment | Tailoring | Change | Risk | Complexity | Strategy | Social | Economic |
|---|---|---|---|---|---|---|---|---|---|---|---|
| | p1 | M | M | VL | H | H | VH | VH | H | M | M |
| DM1 | p2 | M | M | M | M | M | L | M | M | M | EH |
| | p3 | VL | M | M | VL | M | M | H | M | M | M |
| | p1 | M | L | M | H | M | H | VH | M | VH | M |
| DM2 | p2 | M | H | M | H | M | M | M | M | M | M |
| | p3 | M | M | M | M | M | M | M | H | M | M |
| | p1 | M | L | H | EH | EH | EH | EH | H | M | H |
| DM3 | p2 | M | M | H | M | VH | H | EH | H | EH | M |
| | p3 | H | L | H | M | M | M | L | H | M | M |

The weights of the decision-makers are given in Table 5. For this purpose, Equation (11) was used. The weights of decision makers 1 and 3 were equal, and the comments made by these two decision makers were more important than those made by decision maker 2.

**Table 5.** Weights of decision-makers.

| Decision Maker | Linguistic Term | Weight |
|---|---|---|
| DM1 | Very High | 0.34 |
| DM2 | High | 0.32 |
| DM3 | Very High | 0.34 |

The PFWA matrix is presented in Table 6. To address this matrix, Equation (10) was employed in DSS model management. In this equation, $w_j$ refers to the weight of decision-makers. As an example, this table shows that the average opinions of the three decision-makers regarding project 1 were equal to 0.55, 0.45, and 0.704. In this table, the opinions of all the decision-makers about each project, according to the weight of each decision-maker, were merged into one opinion.

**Table 6.** Pythagorean fuzzy weighted averaging matrix (μ, v, π).

| Title | Stakeholders | Value | Environment | Change | Risk |
|---|---|---|---|---|---|
| P1 | (0.55, 0.45, 0.704) | (0.434, 0.573, 0.695) | (0.526, 0.492, 0.694) | (0.723, 0.284, 0.63) | (0.769, 0.234, 0.595) |
| P2 | (0.55, 0.45, 0.704) | (0.586, 0.416, 0.695) | (0.588, 0.413, 0.695) | (0.637, 0.368, 0.677) | (0.541, 0.468, 0.699) |
| P3 | (0.526, 0.492, 0.694) | (0.496, 0.51, 0.703) | (0.588, 0.413, 0.695) | (0.55, 0.45, 0.704) | (0.55, 0.45, 0.704) |
| Title | Complexity | Tailoring | Strategy | Social | Economic |
| P1 | (0.791, 0.21, 0.575) | (0.742, 0.262, 0.617) | (0.622, 0.379, 0.685) | (0.631, 0.374, 0.68) | (0.588, 0.413, 0.695) |
| P2 | (0.701, 0.309, 0.643) | (0.586, 0.416, 0.695) | (0.588, 0.413, 0.695) | (0.701, 0.309, 0.643) | (0.701, 0.309, 0.643) |
| P3 | (0.541, 0.468, 0.699) | (0.478, 0.536, 0.696) | (0.62, 0.381, 0.686) | (0.55, 0.45, 0.704) | (0.55, 0.45, 0.704) |

The normalized Pythagorean fuzzy weighted averaging (NPFWA) matrix is shown in Table 7, and it was calculated using Equation (19). To calculate the score function, Equation (9) was used. For example, the score function of 0.55, 0.45, and 0.704 was equal to 0.605, which is displayed in the first cell of Table 7.

**Table 7.** Normalized Pythagorean fuzzy weighted averaging (μ, v, π) score function.

| Title | Stakeholders | Value | Environment | Change | Risk |
|---|---|---|---|---|---|
| P1 | (0.55, 0.45, 0.704), 0.739 | (0.434, 0.573, 0.695), 0.586 | (0.526, 0.492, 0.694), 0.699 | (0.723, 0.284, 0.63), 1.114 | (0.234, 0.769, 0.595), 0.432 |
| P2 | (0.55, 0.45, 0.704), 0.739 | (0.586, 0.416, 0.695), 0.8 | (0.588, 0.413, 0.695), 0.803 | (0.637, 0.368, 0.677), 0.899 | (0.468, 0.541, 0.699), 0.624 |
| P3 | (0.526, 0.492, 0.694), 0.699 | (0.496, 0.51, 0.703), 0.66 | (0.588, 0.413, 0.695), 0.803 | (0.55, 0.45, 0.704), 0.739 | (0.45, 0.55, 0.704), 0.605 |
| Title | Complexity | Tailoring | Strategy | Social | Economic |
| P1 | (0.21, 0.791, 0.575), 0.42 | (0.742, 0.262, 0.617), 1.173 | (0.622, 0.379, 0.685), 0.868 | (0.631, 0.374, 0.68), 0.885 | (0.588, 0.413, 0.695), 0.803 |
| P2 | (0.309, 0.701, 0.643), 0.476 | (0.586, 0.416, 0.695), 0.8 | (0.588, 0.413, 0.695), 0.803 | (0.701, 0.309, 0.643), 1.051 | (0.701, 0.309, 0.643), 1.051 |
| P3 | (0.468, 0.541, 0.699), 0.624 | (0.478, 0.536, 0.696), 0.635 | (0.62, 0.381, 0.686), 0.864 | (0.55, 0.45, 0.704), 0.739 | (0.55, 0.45, 0.704), 0.739 |

In Table 7, the score function was calculated so that the maximum and minimum could be obtained for the calculation of AAI and AI, which were calculated using Equation (18). The results regarding creating ideal and anti-ideal solutions for the MARCOS method are given in the matrix in Table 8. For example, AI represents the maximum in the column and is the largest for a given value, which is a positive criterion. Among the numbers 0.586, 0.8, and 0.66, the largest number, which is 0.8, was selected, and its fuzzy average is given in Table 8.

The weights of the criteria were calculated using the MEREC method, the normalized cumulative matrix of which is shown in Table 9. This calculation was performed using Equation (13). It was emphasized that normalization for positive and negative criteria in the MEREC method was the opposite of that for the MARCOS method.

**Table 8.** Normal matrix with ideal and anti-ideal solutions.

| Title | Stakeholders | Value | Environment | Change | Risk |
|---|---|---|---|---|---|
| AAI | (0.526, 0.492) | (0.434, 0.573) | (0.526, 0.492) | (0.55, 0.45) | (0.468, 0.541) |
| AI | (0.55, 0.45) | (0.586, 0.416) | (0.588, 0.413) | (0.723, 0.284) | (0.234, 0.769) |
| P1 | (0.55, 0.45) | (0.434, 0.573) | (0.526, 0.492) | (0.723, 0.284) | (0.234, 0.769) |
| P2 | (0.55, 0.45) | (0.586, 0.416) | (0.588, 0.413) | (0.637, 0.368) | (0.468, 0.541) |
| P3 | (0.526, 0.492) | (0.496, 0.51) | (0.588, 0.413) | (0.55, 0.45) | (0.45, 0.55) |
| **Title** | **Complexity** | **Tailoring** | **Strategy** | **Social** | **Economic** |
| AAI | (0.468, 0.541) | (0.478, 0.536) | (0.588, 0.413) | (0.55, 0.45) | (0.55, 0.45) |
| AI | (0.21, 0.791) | (0.742, 0.262) | (0.622, 0.379) | (0.701, 0.309) | (0.701, 0.309) |
| P1 | (0.21, 0.791) | (0.742, 0.262) | (0.622, 0.379) | (0.631, 0.374) | (0.588, 0.413) |
| P2 | (0.309, 0.701) | (0.586, 0.416) | (0.588, 0.413) | (0.701, 0.309) | (0.701, 0.309) |
| P3 | (0.468, 0.541) | (0.478, 0.536) | (0.62, 0.381) | (0.55, 0.45) | (0.55, 0.45) |

**Table 9.** Normal cumulative matrix of MEREC method ($\mu,v,\pi$) score function.

| Title | Stakeholders | Value | Environment | Change | Risk |
|---|---|---|---|---|---|
| P1 | (0.45, 0.55, 0.704), 0.739 | (0.573, 0.434, 0.695), 0.586 | (0.492, 0.526, 0.694), 0.699 | (0.284, 0.723, 0.63), 1.114 | (0.769, 0.234, 0.595), 0.432 |
| P2 | (0.45, 0.55, 0.704), 0.739 | (0.416, 0.586, 0.695), 0.8 | (0.413, 0.588, 0.695), 0.803 | (0.368, 0.637, 0.677), 0.899 | (0.541, 0.468, 0.699), 0.624 |
| P3 | (0.492, 0.526, 0.694), 0.699 | (0.51, 0.496, 0.703), 0.66 | (0.413, 0.588, 0.695), 0.803 | (0.45, 0.55, 0.704), 0.739 | (0.55, 0.45, 0.704), 0.605 |
| **Title** | **Complexity** | **Tailoring** | **Strategy** | **Social** | **Economic** |
| P1 | (0.791, 0.21, 0.575), 0.42 | (0.262, 0.742, 0.617), 1.173 | (0.379, 0.622, 0.685), 0.868 | (0.374, 0.631, 0.68), 0.885 | (0.413, 0.588, 0.695), 0.803 |
| P2 | (0.701, 0.309, 0.643), 0.476 | (0.416, 0.586, 0.695), 0.8 | (0.413, 0.588, 0.695), 0.803 | (0.309, 0.701, 0.643), 1.051 | (0.309, 0.701, 0.643), 1.051 |
| P3 | (0.541, 0.468, 0.699), 0.624 | (0.536, 0.478, 0.696), 0.635 | (0.381, 0.62, 0.686), 0.864 | (0.45, 0.55, 0.704), 0.739 | (0.45, 0.55, 0.704), 0.739 |

The $S_i$ calculations are presented in Table 10. This calculation was conducted using Equation (14). This table summarizes the calculation results of the alternatives' overall performance. The table shows that the overall performance of alternative 2, which was equal to 0.424, was higher than that of the other options.

**Table 10.** $S_i$ calculations.

| Title | $S_i$ |
|---|---|
| P1 | 0.414 |
| P2 | 0.424 |
| P3 | 0.37 |

The $S'_{ij}$ calculations were performed using Equation (15), and the results are presented in Table 11. In this step, by removing each of the criteria, the performance of the alternatives was calculated. In this step, we used the same logarithmic criterion employed in the previous step. The difference between this step and the previous one was that the performance of the alternatives was calculated separately for each criterion. For example, the number

0.38 in the first cell shows the performance of the first alternative after removing the effect of stakeholder criteria.

**Table 11.** $S_{ij}^{/}$ calculations.

| Title | Stakeholders | Value | Environment | Tailoring | Change | Risk | Complexity | Strategy | Social | Economic |
|-------|-------------|-------|-------------|-----------|--------|------|------------|----------|--------|----------|
| p1 | 0.38 | 0.397 | 0.385 | 0.359 | 0.361 | 0.399 | 0.394 | 0.372 | 0.371 | 0.376 |
| p2 | 0.39 | 0.386 | 0.386 | 0.386 | 0.38 | 0.402 | 0.42 | 0.386 | 0.374 | 0.374 |
| p3 | 0.34 | 0.343 | 0.33 | 0.347 | 0.335 | 0.349 | 0.348 | 0.326 | 0.335 | 0.335 |

The $E_j$ calculations were performed using Equation (16), and the obtained results are presented in Table 12. In this step, the effect of removing criterion *j* was calculated based on the values obtained from the previous steps. For example, the number 0.128 in the first cell of the table shows the effect of removing the social criteria.

**Table 12.** $E_j$ calculations.

| Criteria Name | Social | Change | Complexity | Economic | Risk | Environment | Stakeholders | Strategy | Tailoring | Value |
|---------------|--------|--------|------------|----------|------|-------------|--------------|----------|-----------|-------|
| $E_j$ | 0.128 | 0.132 | 0.046 | 0.123 | 0.058 | 0.107 | 0.098 | 0.124 | 0.116 | 0.082 |

The final weights were obtained using Equation (17), and the obtained results are shown in Table 13. This table shows that the criteria change, social, strategy, and economic had the highest weights.

**Table 13.** Final weights.

| Criteria Name | Complexity | Risk | Value | Stakeholders | Environment | Tailoring | Economic | Strategy | Social | Change |
|---------------|------------|------|-------|--------------|-------------|-----------|----------|----------|--------|--------|
| $W_j$ | 0.045 | 0.057 | 0.081 | 0.097 | 0.106 | 0.114 | 0.121 | 0.122 | 0.126 | 0.13 |

The weighted matrix generated by multiplying the weights of the MEREC method by those of the MARCOS cumulative matrix is shown in Table 14. For this purpose, Equation (5) was used.

**Table 14.** Weighted matrix.

| Title | Stakeholders | Value | Environment | Tailoring | Change |
|-------|-------------|-------|-------------|-----------|--------|
| AAI | (0.176, 0.934) | (0.129, 0.956) | (0.184, 0.928) | (0.171, 0.931) | (0.214, 0.901) |
| AI | (0.185, 0.925) | (0.183, 0.931) | (0.21, 0.911) | (0.295, 0.858) | (0.303, 0.849) |
| P1 | (0.185, 0.925) | (0.129, 0.956) | (0.184, 0.928) | (0.295, 0.858) | (0.303, 0.849) |
| P2 | (0.185, 0.925) | (0.183, 0.931) | (0.21, 0.911) | (0.216, 0.905) | (0.256, 0.878) |
| P3 | (0.176, 0.934) | (0.15, 0.947) | (0.21, 0.911) | (0.171, 0.931) | (0.214, 0.901) |
| **Title** | **Risk** | **Complexity** | **Strategy** | **Social** | **Economic** |
| AAI | (0.118, 0.966) | (0.105, 0.973) | (0.225, 0.898) | (0.211, 0.904) | (0.207, 0.908) |
| AI | (0.057, 0.985) | (0.045, 0.99) | (0.241, 0.888) | (0.286, 0.862) | (0.28, 0.868) |
| P1 | (0.057, 0.985) | (0.045, 0.99) | (0.241, 0.888) | (0.249, 0.883) | (0.224, 0.899) |
| P2 | (0.118, 0.966) | (0.067, 0.984) | (0.225, 0.898) | (0.286, 0.862) | (0.28, 0.868) |
| P3 | (0.113, 0.966) | (0.105, 0.973) | (0.24, 0.889) | (0.211, 0.904) | (0.207, 0.908) |

Table 15 shows other elements of the MARCOS method. These calculations were performed using Equations (21)–(26). In the equations, $S_1$ (0.604, 0.411) represents the sum of the rows of the first alternative, $k_1^{+}$ (0.672, 0.264) represents the degree of ideal utility of

the first alternative, and $k_1^-$ (0.716, 0.217) represents the degree of anti-ideal utility of the first alternative, while $f_{k_1}^+$ (0.717, 0.185) and $f_{k_1}^-$ (0.674, 0.226) represent the performance regarding the ideal and anti-ideal utility of the first alternative, respectively.

**Table 15.** Other MARCOS calculations.

| Title | $S_i$ | $k_i^+$ | $k_i^-$ | $f_{k_i}^+$ | $f_{k_i}^-$ | Score Function | | | |
|-------|-------|---------|---------|-------------|-------------|---------|---------|-----------|-----------|
| | | | | | | $k_i^+$ | $k_i^-$ | $f_{k_i}^+$ | $f_{k_i}^-$ |
| P1 | (0.604, 0.411) | (0.672, 0.264) | (0.716, 0.217) | (0.717, 0.185) | (0.674, 0.226) | 0.991 | 1.105 | 1.113 | 1 |
| P2 | (0.611, 0.398) | (0.678, 0.256) | (0.72, 0.21) | (0.721, 0.181) | (0.679, 0.22) | 1.005 | 1.118 | 1.125 | 1.013 |
| P3 | (0.542, 0.464) | (0.625, 0.298) | (0.677, 0.244) | (0.679, 0.2) | (0.628, 0.244) | 0.89 | 1.004 | 1.016 | 0.904 |

The final ranking is presented in Table 16; this ranking was determined using Equation (24). This ranking shows that the second, first, and third projects attained the highest points of 2.423, 2.333, and 1.737, respectively. These ratings showed that mine project No. 2, considering the ten specified criteria, had a higher priority for investment and selection for the holding company. After that, mining projects number 1 and 3 had higher priority. This helped the organization in question to make the right choice and was a tool for the greater success of the organization.

**Table 16.** Final ranking.

| Title | $f_k$ |
|-------|-------|
| P1 | 2.333 |
| P2 | 2.423 |
| P3 | 1.737 |

## 6. Sensitivity Analysis and Discussion of Results

In this section, the effect of slight changes in criteria weights on the final rankings of four different scenarios are examined. In the first state, the weights of criteria 1 and 3 were altered together and those of criteria 8 and 9 were both altered. In the second state, in addition to the changes instituted in the first state, the weight of criterion 22 was replaced by that of 4. In the third state, the weights of criterion 1 were replaced by those of 2, the weights of criterion 3 were replaced by those of 4, those of criteria 5 were replaced by those of 6, and those of 7 were replaced by those of 8. In the fourth state, all criteria were assigned an equal weight of 0.1.

The results obtained from these states are presented in Table 17 and Figure 13, which show that the final ranking was independent from the weights of the criteria. Figure 13 and Table 17 showed that despite the change in the weights in the four different states, the final ranking was not changed. It can be seen in Table 17 that projects 2, 1, and 3 ranked from first to third, respectively.

**Table 17.** Sensitivity analysis of criteria weights.

| Alternatives | Initial Rating | First State | Second State | Third State | Fourth State |
|--------------|----------------|-------------|--------------|-------------|--------------|
| P1 | 2.333 | 2.34 | 2.234 | 2.002 | 2.122 |
| P2 | 2.423 | 2.41 | 2.435 | 2.371 | 2.319 |
| P3 | 1.737 | 1.735 | 1.758 | 1.78 | 1.783 |

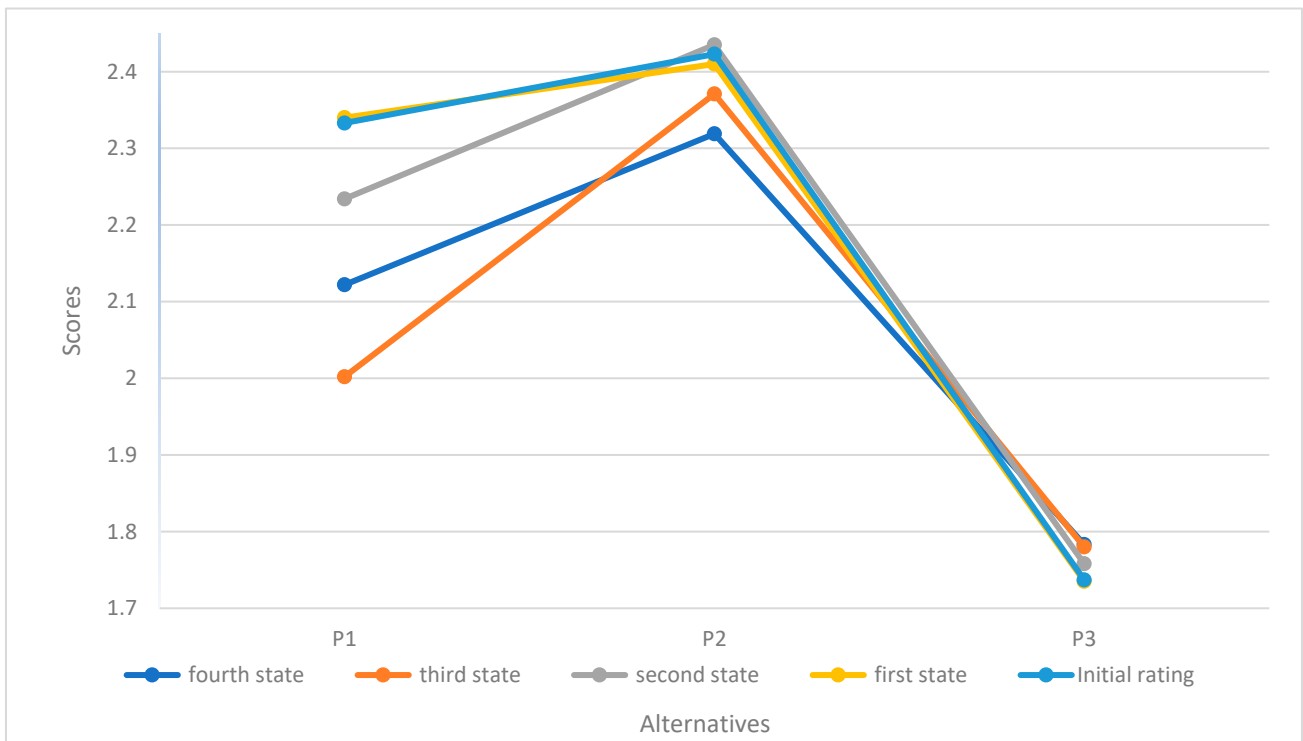

**Figure 13.** Sensitivity analysis of weights.

In the following, a comparison is made between the ranking results obtained using two recently developed PF-Entropy-TOPSIS [59] and PF-Entropy-VIKOR [60] methods as well as the method developed in this research. As shown in Table 18 and Figure 14, the best option for all cases was shown to be P2. In other words, the superior option introduced using the approach presented in this research was the same as that introduced using method 2. After calculating the results, the rankings were presented to the decision makers, who approved them.

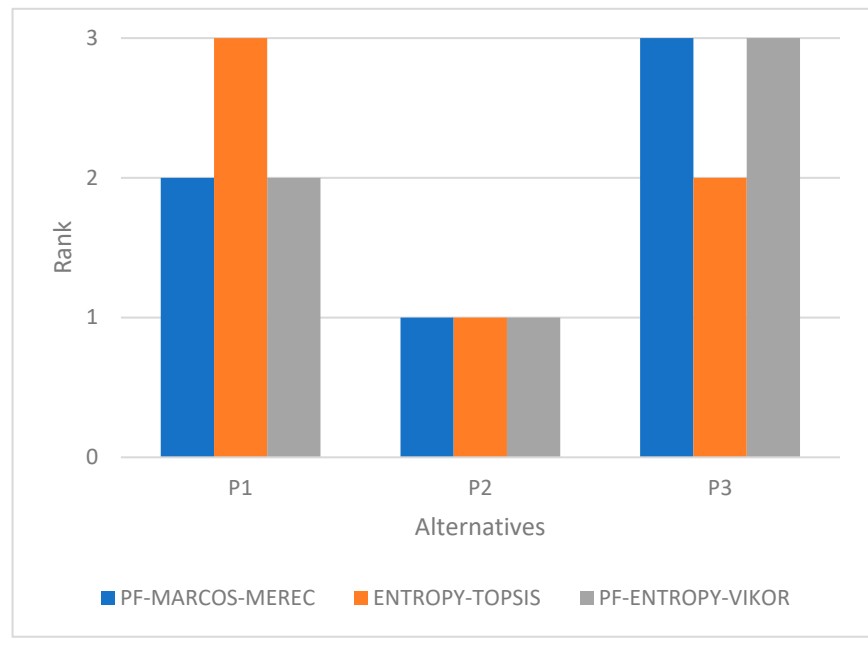

**Figure 14.** Comparison results.

**Table 18.** Comparison of results.

| Alternatives | PF-MEREC-MARCOS | Rank | PF-Entropy-TOPSIS [59] | Rank | PF-Entropy-VIKOR (Q) [59] | Rank |
|:---:|:---:|:---:|:---:|:---:|:---:|:---:|
| P1 | 2.333 | Rank 2 | 0.4258 | Rank 3 | 0.561 | Rank 2 |
| P2 | 2.423 | Rank 1 | 0.589 | Rank 1 | 0 | Rank 1 |
| P3 | 1.737 | Rank 3 | 0.5581 | Rank 2 | 0.701 | Rank 3 |

## 7. Managerial Implications

Although a potential new methodology has been proposed in this research so far, determining how to realize this methodology is also important. This task requires prerequisites that depend on system and human factors. The system part refers to the essential issues that have to be addressed for this system and approach to work correctly, and the human part refers to the tasks people related to this system have to perform for the system to work correctly. Our view on the further achievement of this methodology in terms of the system and human parts includes the following:

- Creating a consensus in the company: This means that the decision-makers in the company reach a consensus that the current process is ineffective and the company needs a better and more social process to select workable projects.
- Agreeing on methodological principles: These principles include the weights of decision-makers, the number of criteria, and selection criteria. Although the selection methodology is flexible, can be developed for multiple criteria, and assign different weights to the decision makers, for the realization of the above methodology, an agreement is required on the current principles.
- Human aspect: From this point of view, the decision-makers who use the above system should have received the necessary training to use the current system and must also be able to make changes in the system to develop new items.

## 8. Conclusions

The issue of choosing projects in organizations, especially holding companies, has become a fundamental challenge. It is possible to choose a project for an organization using political and managerial approaches. Such an approach does not ensure that the organization's objectives will be met, and the senior managers of the organization should focus on determining the frameworks and goals of their organization. A system analyst can support the decision-making process by applying thorough analysis under these circumstances. Using multi-criteria decision making and combining it with decision support systems is an approach that helps senior managers of organizations solve project selection problems. In this article, such an approach has been proposed and applied to a mineral holding company in Iran. Pythagorean fuzzy sets (PFSs) are an extension of intuitionistic fuzzy sets. These sets increase the capacity of decision-makers (DMs) to express their ideas. These sets provide more flexibility for DMs and hence can lead to better handling of real decision problems. This paper sought to propose a new web-based decision support system by presenting two new versions of PFS and MARCOS-MEREC approaches. The novelty of these approaches, the development of MARCOS-MEREC methods with Pythagorean fuzzy sets, the possibility of considering a large number of criteria and alternatives, the simplicity of calculations, and the comprehensibility of the methods and their high stability are some of the advantages of the approach introduced in this research.

In other words, in this approach, a new extension to the MEREC and MARCOS methods was made using Pythagorean fuzzy sets and DSS. Considering the online accessibility of the internet in today's societies, providing a web-based decision support system to evaluate projects may be one of the most important aims of this paper. One of the significant contributions of the current research regarding project evaluation indicators and criteria is its consideration of the principles of PMBOK along with sustainable development in the desired criteria and indicators for the project review. The use of score functions combined with the degree of uncertainty in the PF-MEREC method is another innova-

tive contribution of the current research. Performing fuzzy calculations using the latest Pythagorean fuzzy operators until the final steps of the MARCOS method and postponing defuzzification until the final step of this method are the groundbreaking features of this method that few researchers have explored, through which the accuracy of calculations increased enormously. The score functions used in the current study also took into account the degree of uncertainty. A case study on one of the mineral holdings was examined to demonstrate the applicability of this approach. In this study, three mining projects in a real case study were examined in terms of 10 criteria, most of which were selected from the 7th edition of the *Project Management Standard and Dimensions of Sustainability Development*, and finally, projects 1, 3, and 2 were evaluated as being suitable for investment. Through the comparison made with the well-known PF-Entropy-TOPSIS and PF-Entropy-VIKOR methods, the top ranking was confirmed. Also, with the sensitivity analysis performed on the weights of the criteria, it was determined that the results of weight change scenarios did not affect the final results.

Future researchers are encouraged to use other Pythagorean sets to develop and improve the developed approach, as this can improve the flexibility of the approach in evaluating the applicability of PFSs in the real-world. The proposed approach can be improved with different types of extended fuzzy set methods [61], like hesitant fuzzy sets, to deal with complex uncertain situations. Using interval-valued PFSs to extend the method could be the way forward. Incorporating other MCDM techniques into the proposed decision support system is another avenue for future research. The use of other weighting techniques as well as a combination of different weighting techniques can be considered as other directions that can lead to the development of the above decision support system. In addition, another idea for future research is to compare the outcomes of alternative procedures with those of the current study. Also, the proposed method can be employed in all other industries. In this research, it is assumed that the decision-makers are independent from each other. Another future suggestion can be considering the dependence among decision-makers, including the social networks among them. Considering the dependence between criteria can be another topic that can be developed with new extensions of fuzzy sets [62,63]. Furthermore, adding more criteria of sustainable development to evaluate projects is another approach that can be explored by researchers.

**Author Contributions:** Conceptualization, data curation, methodology, and writing—original draft preparation, A.M.A.S.; methodology, validation, supervision, writing—review and editing, S.M.M. All authors have read and agreed to the published version of the manuscript.

**Funding:** This research received no external funding.

**Institutional Review Board Statement:** Not applicable.

**Informed Consent Statement:** Not applicable.

**Data Availability Statement:** The data used in this paper are available on request from the corresponding author.

**Conflicts of Interest:** The authors declare no conflict of interest.

## Appendix A

In this part, Figure A1 shows the function related to the multiplication operator, Figure A2 shows the function related to the division operator, and Figure A3 depicts the function related to the subtraction operator.

```sql
USE [DSS]
GO
/****** Object:  UserDefinedFunction [dbo].[FLMultiple]    Script Date: 01/11/1401 02:27:58    ******/
SET ANSI_NULLS ON
GO
SET QUOTED_IDENTIFIER ON
GO

ALTER FUNCTION [dbo].[FLMultiple](@Landa float,@A nvarchar(100))

 RETURNS nvarchar(100) AS
 BEGIN
     declare @S nvarchar(100)

     declare @M nvarchar(100)
     declare @V nvarchar(100)

     set @M=dbo.PartA(@A)
     set @V=dbo.PartB(@A)

     set @S= '( ' + cast (round(sqrt(1-POWER( 1-Power(@M,2),@Landa)),3)  as nvarchar)  + ', ' + cast(round(power(@V,@Landa),3)  as nvarchar) + ' )'

   return (@S)
 END
```

**Figure A1.** Function related to the multiplication operator in DSS.

```sql
USE [DSS]
GO
/****** Object:  UserDefinedFunction [dbo].[FDivision]    Script Date: 01/11/1401 02:30:16    ******/
SET ANSI_NULLS ON
GO
SET QUOTED_IDENTIFIER ON
GO

ALTER FUNCTION [dbo].[FDivision](@A1 nvarchar(100),@A2 nvarchar(100))

 RETURNS nvarchar(100) AS
 BEGIN
     declare @S nvarchar(100)

     declare @M1 float
     declare @V1 float
     declare @M2 float
     declare @V2 float

     set @M1=cast(dbo.PartA(@A1) as float)
     set @V1=cast(dbo.PartB(@A1) as float)

     set @M2=cast(dbo.PartA(@A2) as float)
     set @V2=cast(dbo.PartB(@A2) as float)

     set @S= '( ' + cast(sqrt(power(@M1,2)+power(@V2,2)-(power(@M1,2)*power(@V2,2))) as nvarchar) + ' , ' + cast(@V1 * @M2  as nvarchar)  + ' )'

   return (@S)
 END
```

**Figure A2.** Function related to the division operator in DSS.

```
USE [DSS]
GO
/****** Object:  UserDefinedFunction [dbo].[FMinus]    Script Date: 01/11/1401 02:29:40    ******/
SET ANSI_NULLS ON
GO
SET QUOTED_IDENTIFIER ON
GO

ALTER FUNCTION [dbo].[FMinus](@A1 nvarchar(100),@A2 nvarchar(100))

RETURNS nvarchar(100) AS
BEGIN
    declare @S nvarchar(100)

    declare @M1 nvarchar(100)
    declare @V1 nvarchar(100)
    declare @M2 nvarchar(100)
    declare @V2 nvarchar(100)

    set @M1=dbo.PartA(@A1)
    set @V1=dbo.PartB(@A1)

    set @M2=dbo.PartA(@A2)
    set @V2=dbo.PartB(@A2)

    set @S= '( ' + cast(cast(@M1 as float)*cast(@V2 as float) as nvarchar) + ' , ' + cast(sqrt(power(@V1,2)+power(@M2,2)-(power(@V1,2)*power(@M2,2))) as nvarchar)  + ' )'

    return (@S)
END
```

**Figure A3.** Function related to the subtraction operator in DSS.

Another function used in this DSS is related to score functions, as shown in Figure A4.

```
USE [DSS]
GO
/****** Object:  UserDefinedFunction [dbo].[ScoredFunction1]    Script Date: 01/11/1401 02:31:11    ******/
SET ANSI_NULLS ON
GO
SET QUOTED_IDENTIFIER ON
GO

ALTER FUNCTION [dbo].[ScoredFunction1](@A nvarchar(100))

RETURNS float AS
BEGIN
    declare @S float

    declare @M float
    declare @V nvarchar(100)

    set @M=cast( dbo.PartA(@A) as float)
    set @V=cast( dbo.PartB(@A) as float)

    set @S=round( EXP(Square(@M) - Square(@V)) / (1 + Square(dbo.Hesitant(@A))),3)

    return (@S)
END
```

**Figure A4.** Score function in DSS.

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
