# Peer review of "A Web-Based Decision Support System for Project Evaluation with Sustainable Development Considerations Based on Two Developed Pythagorean Fuzzy Decision Methods"

_sustainability, doi:10.3390/su152316477_

Round 1

Reviewer 1 Report

Comments and Suggestions for Authors

The concept is very interesting. The paper is very comprehensive with background information well-covered and significant added knowledge.

Few comments for authors to better improve the manuscript and make it ready for publishing is:

Please revise the manuscript’s structure and organization, the introduction is long, while some of its parts can be moved to a literature review section.

Table 1 can be moved to the method section

Section 2 is entitled ‘preliminary’ which is not clear, please revise

Please consider deleting figure 3

Too many figures, tables and equations, please consider merging some of them

Comments on the Quality of English Language

minor revision required

Author Response

Reviewer #1:

The concept is very interesting. The paper is very comprehensive with background information well-covered and significant added knowledge.

Few comments for authors to better improve the manuscript and make it ready for publishing is:

Comment 1. Please revise the manuscript's structure and organization, the introduction is long, while some of its parts can be moved to a literature review section.

Respond 1. Thanks for your comment. The number of articles, which was regarded as this paper's main topic, was limited. Hence, we did not separate the literature review from the introduction section. Due to the multitude of concepts in the paper, including Pythagorean fuzzy, decision support systems, and MCDM approaches such as MARCOS and MEREC methods, an effort has been made to review and investigate each of these concepts more carefully.

Comment 2. Table 1 can be moved to the method section

Respond 2. Thanks for your comment. This table has an essential role in presenting the role of this paper by comparing the previous articles and analyzing the gap in the related literature. According to this issue, table 1 can help to examine the gap of the studies, and this issue was caused to use this table in the section. This table has a fundamental role in evaluating the gap of the previous papers and demonstrates the strength points and innovations of the proposed paper.  

Comment 3. Section 2 is entitled 'preliminary' which is not clear, please revise

Respond 3. Thanks for your comment. In this section, basic definitions of PFS and operators of PFS were presented. Some items were added to clarify this section, and the above changes were highlighted on pages 6 to 8. For this purpose, more explanations were added to Figure 1, highlighted on page 6. Also, to clarify the topics, the presentation structure of preliminaries was changed in the form of definitions, which were highlighted on pages 7 to 8. Eventually, these definitions were essential for the proposed approach that were provided in this section entirely.

Comment 4. Please consider deleting figure 3

Respond 4. Thanks for your comment. We have removed Figure 3. Furthermore, this figure was changed to the structure of the proposed approach based on the other reviewer that was added and highlighted on page 10.

Comment 5. Too many figures, tables and equations, please consider merging some of them

Respond 5. Thanks for your comment. We checked and changed it completely. Except for Figure 3, which was deleted, all the figures and formulas were checked. All the formulas mentioned in the paper have been used. In this section, we removed Figure 3 and also moved Figures 7- 10 to Appendix A. This area was completely modified, and we checked the tables and figures. Appendix A was highlighted on page 32.

Reviewer 2 Report

Comments and Suggestions for Authors

Report

reviewer comment

A new webbased decision support system for project evaluation

with sustainable development considerations based on

two developed Pythagorean fuzzy decision methods

The current research has developed a new approach using the Pythagorean

fuzzy sets, MEREC, and MARCOS methods to examine uncertainty and the

solution method. In this approach, a new version of MARCOS method, which

is for ranking, is developed with Pythagorean fuzzy sets. Also, a new de-

velopment with Pythagorean fuzzy-MEREC method, which is used for the

weighting, has been presented

There are some minor remarks to the presentation.

1. The authors are suggested to apply proofs of your paper in actual prob-

lem (some examples).

2. all symbols should be italic in math form.

3. All the notations and abbreviations should be checked carefully.

4. Please, rewrite abstract section again.

5. Mention at least one application with elaborate solution .

6. Are these new results sharp and more accurate compared with the others?

7. Abstract should be improved, as it has some shortcomings regarding the

continuity of sentences. It should be revised in the sense that it clearly

present the proposed idea e ectively.

8. Please check and if needed add all the missing punctuation or missing

italics.

9. To increase the interest of the authors in this research, I suggest adding

some applications to the results

10. Theorem is not clear. Needs more discussion.

11. Improve literature review, historical background and add future scope

of study more e ectively. To handle this issue, author can discuss some

well related and recent references.

2

12. Language quality needs improvement.

13. Conclusion section should be enrich with some future remarks.

14. please clearly describe the contributions of your paper in the Abstract.

15. The references should have a clear and uni ed format .

16. Please cite more papers in your paper to catch current signi cant devel-

opments Pythagorean fuzzy sets.

17. it contains many shortcomings in the language, especially taking into

account the inde nite letters a, an, the ..etc. and unifying its use in all

paper

18. I have reviewed the paper. Presentation, the related literature and tech-

nical soundness are weak. It presents good method. After made some

additions to the paper, i recommend for publication

19. I suggest adding some new relevant references to this topic.

(a) On Mixed b-Fuzzy Topological Spaces, 2020,International Journal

of Fuzzy Logic and Intelligent Systems, Volume (3),242-246.

(b) Multi granulation on nano soft topological space, Advances in Math-

ematics: Scienti c Journal, 2020, 9(10), pp. 77117717

Recommendation. I recommend accepting and publishing the paper after tak-

ing all the previous comments and observations and making some minor mod-

i cations mentioned above

Received: Month 9, day 9

Comments on the Quality of English Language

1. The authors are suggested to apply proofs of your paper in actual prob-

lem (some examples).

2. all symbols should be italic in math form.

3. All the notations and abbreviations should be checked carefully.

4. Please, rewrite abstract section again.

5. Mention at least one application with elaborate solution .

6. Are these new results sharp and more accurate compared with the others?

7. Abstract should be improved, as it has some shortcomings regarding the

continuity of sentences. It should be revised in the sense that it clearly

present the proposed idea e ectively.

8. Please check and if needed add all the missing punctuation or missing

italics.

9. To increase the interest of the authors in this research, I suggest adding

some applications to the results

10. Theorem is not clear. Needs more discussion.

11. Improve literature review, historical background and add future scope

of study more e ectively. To handle this issue, author can discuss some

well related and recent references.

2

12. Language quality needs improvement.

13. Conclusion section should be enrich with some future remarks.

14. please clearly describe the contributions of your paper in the Abstract.

15. The references should have a clear and uni ed format .

16. Please cite more papers in your paper to catch current signi cant devel-

opments Pythagorean fuzzy sets.

17. it contains many shortcomings in the language, especially taking into

account the inde nite letters a, an, the ..etc. and unifying its use in all

paper

18. I have reviewed the paper. Presentation, the related literature and tech-

nical soundness are weak. It presents good method. After made some

additions to the paper, i recommend for publication

19. I suggest adding some new relevant references to this topic.

(a) On Mixed b-Fuzzy Topological Spaces, 2020,International Journal

of Fuzzy Logic and Intelligent Systems, Volume (3),242-246.

(b) Multi granulation on nano soft topological space, Advances in Math-

ematics: Scienti c Journal, 2020, 9(10), pp. 77117717

Recommendation. I recommend accepting and publishing the paper after tak-

ing all the previous comments and observations and making some minor mod-

i cations mentioned above

Author Response

Reviewer #2:

Report

reviewer comment

A new webbased decision support system for project evaluation with sustainable development considerations based on two developed Pythagorean fuzzy decision methods

The current research has developed a new approach using the Pythagorean fuzzy sets, MEREC, and MARCOS methods to examine uncertainty and the solution method. In this approach, a new version of MARCOS method, which is for ranking, is developed with Pythagorean fuzzy sets. Also, a new development with Pythagorean fuzzy-MEREC method, which is used for the weighting, has been presented

There are some minor remarks to the presentation.

Comment 1. The authors are suggested to apply proofs of your paper in actual problem (some examples).

Respond 1. Thanks for your comment. We have added the full description of the real case study in this paper . This was highlighted on pages 16 and 17. Also, the presented case study was actual and was approved by the experts. Since the mineral company was looking for a systematic solution to evaluate the candidate projects, the decision support system provided in this paper was welcomed, and this system was used in a completely real environment; also, its results were approved by the company's experts.

Comment 2. all symbols should be italic in math form.

Respond 2. Thanks for your comment. This was done. These main modifications were highlighted in the preliminary section. These items were highlighted on pages 6 to 8 and 12 to 13.

Comment 3. All the notations and abbreviations should be checked carefully.

Respond 3. Thanks for your comment. A general and complete review was done, and the necessary modifications were made; an example of these modifications was highlighted on page 6 of the last paragraph.

Comment 4. Please, rewrite abstract section again.

Respond 4. Thanks for your valuable mention. Modifications were given and highlighted in the abstract. In addition to making more connections between the sentences in the abstract, an effort was made to refer more to the dimensions of sustainable development. The result of the implementation of the case study was added. Also, the comparison of the proposed methodology with two other methods was added to this section. We also added more explanations about sensitivity analysis. All these modifications were highlighted on page 1.

Comment 5. Mention at least one application with elaborate solution.

Respond 5. Thanks for your comment. We conducted a real case study with a fully functional case. In mineral holding for the purpose of applicability, the application of the proposed method was used by the company's decision makers, and this decision support system was helpful as a basis for choosing the project for applicability. Also, in order to highlight the solution and make the connection between tables and formulas clearer, modifications were made in the form of highlights on pages 2, 15, 19 - 26.

Comment 6. Are these new results sharp and more accurate compared with the others?

Respond 6. Thanks for your comment. In order to check the accuracy of the proposed method, another comparison was added to the research, and in that the proposed method was compared to PF-Entropy-TOPSIS and PF-Entropy-VIKOR, the same results were obtained. The results and comparison of the added method with the proposed methodology were highlighted on pages 28 and 29 in Table 18 and Figure 14.

Comment 7. Abstract should be improved, as it has some shortcomings regarding the continuity of sentences. It should be revised in the sense that it clearly present the proposed idea effectively.

Respond 7. Thanks for your mention. Modifications were made in the abstract, and efforts were made to include more comprehensiveness. In addition to making more connections between the sentences in the abstract, an effort was made to refer more to the dimensions of sustainable development. The result of the implementation of the case study was added. Also, the comparison of the proposed methodology with two other methods was added to this section. We also added more explanations about sensitivity analysis. All these modifications were highlighted on page 1.

Comment 8. Please check and if needed add all the missing punctuation or missing italics.

Respond 8. Thanks for your comment. A comprehensive review was done throughout the revised paper. In addition to the extensive improvement in punctuation, non-italic items were modified, an example of which was highlighted on pages 6-8 and 12&13.

Comment 9. To increase the interest of the authors in this research, I suggest adding some applications to the results

Respond 9. Thanks for your mention. We provided a real case study and solved it with the proposed methodology. We analyzed different sensitivities in the field of weight in section 6 and compared the results. The comparison with one method was upgraded to the comparison with 2 methods, and these modifications were highlighted on pages 28 and 29. The results were confirmed by experts who were hired for this project. In addition, our decision support system has the ability to be customized for different industries, with different criteria, the number of multiple projects, and the number of multiple decision makers. We think that these features can be added to the attractiveness of its use. In addition, its most important feature was being on the web, which makes it available to everyone at any time and place through the Internet.

Comment 10. Theorem is not clear. Needs more discussion.

Respond 10. Thanks for your comment. More background was added in the proposed soft computing model section. In particular, some modifications were added in sections 3 and 4, which were highlighted. A flowchart was added in order to clarify the step-by-step implementation of the steps on page 10 after Figure 3. Also, on the same page, more explanations were added to the criteria section, which were highlighted. Another change that helped clarify the theory was the clarification of equation 13 on page 11, which was updated from crisp to fuzzy, which can significantly help readers. These changes were evident on pages 8,10,11,14 and 16.

Comment 11. Improve literature review, historical background and add future scope of study more effectively. To handle this issue, author can discuss some well related and recent references.

Respond 11. Thanks for your mention. Improvements were added to the literature review, which were particularly evident on page 5 and in the literature review table. Several articles were added on pages 2, 3 and 4. New articles were added and highlighted in Table 1 on page 5. Also, these modifications led to extensive rewriting in the paragraph after Table 1, which were highlighted on page 5. Also, in order to further strengthen this section, research questions were added as highlights on page 6.

Comment 12. Language quality needs improvement.

Respond 12. Thanks for your valuable mention. The quality of whole revised writing was reviewed carefully by an English expert.

Comment 13. Conclusion section should be enrich with some future remarks.

Respond 13. Thanks for your comment. More suggestions for future research were added, which were highlighted in the conclusion section. These suggestions were highlighted on pages 30-31.

Comment 14. please clearly describe the contributions of your paper in the abstract.

Respond 14. Thanks for your comment. Modifications and corrections were made in the abstract. In addition to making more connections between the sentences in the abstract, an effort was made to refer more to the dimensions of sustainable development. The result of the implementation of the case study was added. Also, the comparison of the proposed methodology with two other methods was added to this section. We also added more explanations about sensitivity analysis. All these modifications were highlighted on page 1.

Comment 15. The references should have a clear and unied format .

Respond 15. Thanks for your mention. References were investigated, and necessary measures were taken. Among these matters was adding DOI to the references that lacked it. In this way, all sources were completed with DOI.

Comment 16. Please cite more papers in your paper to catch current signicant developments Pythagorean fuzzy sets.

Respond 16. Thanks for your comment. More articles related to PFSs were reviewed, and this was added and highlighted on page 4. Several articles in the fuzzy field were added on pages 3 and 4 and in Table 1.

Comment 17. it contains many shortcomings in the language, especially taking into account the indenite letters a, an, the ..etc. and unifying its use in all paper

Respond 17. Thanks for your mention. The whole revised paper was improved by an English expert.

Comment 18. I have reviewed the paper. Presentation, the related literature and technical soundness are weak. It presents good method. After made some additions to the paper, i recommend for publication

Respond 18. Thanks for your comment. Many changes were made in the revised paper. A particular focus was strengthening the introduction section on pages 2 to 6. These modifications were highlighted. Adding research questions and comparisons with more recent articles focusing on Pythagorean fuzzy and MARCOS and MEREC have helped to enrich the literature.

Comment 19. I suggest adding some new relevant references to this topic.

(a) On Mixed b-Fuzzy Topological Spaces, 2020,International Journal

of Fuzzy Logic and Intelligent Systems, Volume (3),242-246.

(b) Multi granulation on nano soft topological space, Advances in Mathematics: Scienti c Journal, 2020, 9(10), pp. 77117717

Recommendation. I recommend accepting and publishing the paper after taking all the previous comments and observations and making some minor modications mentioned above

Respond 19. Thanks for your mention. The above papers were used and highlighted on pages 2 and 3.

Reviewer 3 Report

Comments and Suggestions for Authors

This paper proposes A new web‐based decision support system for project evaluation with sustainable development considerations based on two developed Pythagorean fuzzy decision methods. The paper is well-written and provides valuable insights, but some concerns should be addressed:

• The title of the paper needs refinement to accurately capture its content in a concise and engaging manner. The authors should aim to propose a title that effectively encapsulates the key research findings.

• The abstract should succinctly outline the significant findings and maintain a clear and organized structure. It should offer a brief overview of the primary discoveries to aid reader comprehension.

• The introduction should establish a logical framework for the research, ensuring smooth transitions between paragraphs. The authors should enhance coherence and establish stronger connections between ideas. Additionally, the introduction must explicitly highlight the research's contributions, clarifying the research question and its significance.

• Figures should be presented in a clear and concise manner, accompanied by informative captions to facilitate reader comprehension.

• The research problem should be stated in a clear and concise manner to enhance readability.

• The authors should expound on the connections between the forecasting and Data Augmentation methods, elucidating the contribution of each method to the research. A detailed explanation of the solution method employed and its effectiveness is also warranted.

• Figures and charts should be of high quality, easily legible, visually appealing, and proficient in conveying the research findings. Figure 1 should be revised to incorporate additional data or more detailed information.

• The authors should offer deeper managerial insights derived from the research findings and explicitly articulate the practical implications for managers and organizations.

• The authors should meticulously proofread and edit the paper to rectify any typos or grammatical errors.

Comments on the Quality of English Language

 Extensive editing of English language required

Author Response

Reviewer #3:

This paper proposes A new web‐based decision support system for project evaluation with sustainable development considerations based on two developed Pythagorean fuzzy decision methods. The paper is well-written and provides valuable insights, but some concerns should be addressed: 

Comment 1. The title of the paper needs refinement to accurately capture its content in a concise and engaging manner. The authors should aim to propose a title that effectively encapsulates the key research findings.

Respond 1. Thanks for your mention. The title of the paper has been corrected. For this purpose, MARCOS and MEREC methods were added. According to the opinion of another respected reviewer who wanted to summarize the title, this was done.

Comment 2. The abstract should succinctly outline the significant findings and maintain a clear and organized structure. It should offer a brief overview of the primary discoveries to aid reader comprehension

Respond 2. Thanks for your mention. Modifications were made in the abstract, and efforts were made to include more comprehensiveness. In addition to making more connections between the sentences in the abstract, an effort was made to refer more to the dimensions of sustainable development. The result of the implementation of the case study was added. Also, the comparison of the proposed methodology with two other methods was added to this section. We also added more explanations about sensitivity analysis. All these modifications were highlighted on page 1.

Comment 3. The introduction should establish a logical framepaper for the research, ensuring smooth transitions between paragraphs. The authors should enhance coherence and establish stronger connections between ideas. Additionally, the introduction must explicitly highlight the research's contributions, clarifying the research question and its significance.

Respond 3. Thanks for your comment. Regarding the communication between the contents of the considered reforms, the example was highlighted on pages 2, 14, and 15. Research questions were also added. These items were highlighted on page 6. Also, advantages related to MARCOS and MEREC methods were added on page 2. In order to clarify the comparison of this paper with other studies, the paragraph after Table 1 was rewritten. This change was highlighted on page 5.

Comment 4. Figures should be presented in a clear and concise manner, accompanied by informative captions to facilitate reader comprehension.

Respond 4. Thanks for your comment. The quality of all figures was improved. More explanations were added to the figures, examples of which were highlighted in Figures 1,3,6,8,9,10  and 11. Also, detailed explanations for Figure 3 were added. These modifications are highlighted on pages 6, 10, 16, 17 and 19.

Comment 5. The research problem should be stated in a clear and concise manner to enhance readability.

Respond 5. Thanks for your comment. Explanations were added on pages 10, 14, 15 and 16. Regarding making the research problem clearer, on page 5 after Table 1, the difference between this paper and the previous papers was mentioned. The comprehensiveness of this paper by simultaneously including MCDM, uncertainty, web-based DSS, focusing on the principles of project management version 7, and dimensions of sustainable development for identifying criteria was the most important strength of this paper. An attempt was made to clarify the process by presenting a flowchart in Figure 3. The steps of the problem were explained step by step. In this flowchart, which was determined on page 10,

Comment 6. The authors should expound on the connections between the forecasting and Data Augmentation methods, elucidating the contribution of each method to the research. A detailed explanation of the solution method employed and its effectiveness is also warranted.

Respond 6. Thanks for your comment. The advantages of MARCOS and MEREC methods were highlighted on page 2 and added to the paper. Some formulas were revised, such as formula 13 on page 11. Regarding the further explanation of our methods and paper, these differences were clearly highlighted and expressed on page 5. Also, in order to clarify the paper process, a flowchart was added on page 10.

Comment 7. Figures and charts should be of high quality, easily legible, visually appealing, and proficient in conveying the research findings. Figure 1 should be revised to incorporate additional data or more detailed information.

Respond 7. Thanks for your mention. More explanations were added to the figures, examples of which were highlighted in Figures 1,3,6,8,11 and 12. Also, detailed explanations for Figure 3 were added. These modifications were highlighted on pages 6, 10, 16, 17, 19. The quality of figures was improved. This was evident in Figures 5, 8, 9, and 10.

Comment 8. The authors should offer deeper managerial insights derived from the research findings and explicitly articulate the practical implications for managers and organizations.

Respond 8. Thanks for your mention. Managerial implications were added in section 7. This was highlighted on page 29. In this section, suggestions have been presented from two perspectives: system and human.

Comment 9. The authors should meticulously proofread and edit the paper to rectify any typos or grammatical errors.

Respond 9. The whole revised paper was improved by an English expert.

Reviewer 4 Report

Comments and Suggestions for Authors

The paper proposes a new  web-based decision support system for project evaluation based on two developed Pythagorean fuzzy decision methods however, I do not see any linkages with sustainability in this paper. The case study on mining project is provided with the application of this new method but it is not clear what kind of sustainability indicators where applied. No background for indicator selection. No discussion of indicators. No results, discussion and explanations. Therefore this is paper not about sustainability. It fits better to Mathematics or Algorithms.  Therefore, I am rejecting it as it is fully out of the scope of the journal.

Author Response

Reviewer #4:

The paper proposes a new  web-based decision support system for project evaluation based on two developed Pythagorean fuzzy decision methods however, I do not see any linkages with sustainability in this paper. The case study on mining project is provided with the application of this new method but it is not clear what kind of sustainability indicators where applied. No background for indicator selection. No discussion of indicators. No results, discussion and explanations. Therefore this is paper not about sustainability. It fits better to Mathematics or Algorithms. Therefore, I am rejecting it as it is fully out of the scope of the journal.

Respond. Thanks for your mention. More explanations about sustainability were added in the abstract of the paper and on page 10. These modifications were highlighted. The connection of our research with the field of sustainability was to provide a new approach for researchers, which can be evaluated by looking at the sustainable development of projects. Including criteria related to sustainable development in project evaluation criteria and integrating project evaluation in addition to PMBOK's specialized criteria with the view of sustainable development was the approach used in this research. This web-based support system has the necessary flexibility to add more criteria for sustainable development. Evaluation of projects with three main dimensions of sustainable development has been done in this research.

Reviewer 5 Report

Comments and Suggestions for Authors

This article organizes satisfactorily. However, some graphical interpretations need to develop more, also for clear view their re-production are welcome.

Comments on the Quality of English Language

Good.

Author Response

Reviewer #5:

This article organizes satisfactorily. However, some graphical interpretations need to develop more, also for clear view their re-production are welcome.

Respond. Thanks for your mention. More explanations were added to the figures, examples of which were highlighted in Figures 1,3,6,8,11 and 12. Also, detailed explanations for Figure 3 were added. These modifications were highlighted on pages 6, 10, 16, 17, 19. The quality of figures was improved. This was evident in Figures 5, 8, 9, and 10.

Reviewer 6 Report

Comments and Suggestions for Authors

1. Introduction the main contributions of the work more clearly. The difference between present work and previous. Works should be highlighted.

2. In section 1, more information should be provided for history of fuzzy set and applications, history of intuitionistic fuzzy set and applications.

3.Furthermore, authors should add definitions of fuzzy set, intuitionistic fuzzy set in Section 2.

4. Also, authors should add definitions of Pythagorean fuzzy set and properties  instead of add explain in Section 2.

5. Comparison with recent study and methods (especially with PFS-MEREC methods and PFS-MACROS instead of TOPSIS method) would be appreciated. What is the motivation of the proposed work. Also, Sensitivity analysis and discussion of results are very short.

6. Authors should add especially more PFS-Merec methods and PFS-MACROS in Table 1.

7. Authors should add these references for fuzzy set theory, PFS theory and applications:

- Gülşen, K. U. M., SÖNMEZ, M. E., & KARGIN, A. (2022). An Alternative Process for Determining Erosion Risk: The Fuzzy Method. Coğrafya Dergisi, (44), 219-229.

- Mishra, A. R., Rani, P., Pamucar, D., & Saha, A. (2023). An integrated Pythagorean fuzzy fairly operator-based MARCOS method for solving the sustainable circular supplier selection problem. Annals of Operations Research, 1-42.

- Ali, J. (2022). A q-rung orthopair fuzzy MARCOS method using novel score function and its application to solid waste management. Applied Intelligence52(8), 8770-8792..

- Fan, J., Wang, M., & Wu, M. An extended MEREC-EDAS approach with linguistic pythagorean fuzzy set for selecting virtual team members. Journal of Intelligent & Fuzzy Systems, (Preprint), 1-21.

Author Response

Reviewer #6:

Comments and Suggestions for Authors:

Comment 1. Introduction the main contributions of the paper more clearly. The difference between present paper and previous. Papers should be highlighted.

Respond 1. Thanks for your comment. A complete review was done in the introduction section, and corrections were made; These modifications were highlighted. Regarding the communication between the contents of the considered reforms, the example was highlighted on pages 2,14 and 15. Research questions were also added. These items were highlighted on pages 6. Also, advantages related to MARCOS and MEREC methods were added on page 2. In order to clarify the comparison of this paper with other studies, the paragraph after Table 1 was rewritten and completed. This change was highlighted on page 5. Also, regarding the difference between the current paper and the past paper, detailed explanations were highlighted in a paragraph on page 5.

Comment 2. In section 1, more information should be provided for history of fuzzy set and applications, history of intuitionistic fuzzy set and applications.

Respond 2. Thanks for your comment. More explanations and references were added and highlighted on pages 4 and 5. This was completed and strengthened in Table 1 by adding three references from 2023 that developed MARCOS or MEREC methods with the Pythagorean fuzzy set.

Comment 3. Furthermore, authors should add definitions of fuzzy set, intuitionistic fuzzy set in Section 2.

Respond 3. Thanks for your comment. The definitions of the fuzzy used in our proposed method were given in this section. Modifications were made in the presentation and structure of the definitions, which were highlighted on page 7. In this section, basic definitions of PFS and operators of PFS were presented. Some items were added in order to make this section clearer, and the above modifications were highlighted. On pages 6 to 8, these modifications were highlighted. For this purpose, more explanations were added to Figure 1, which were highlighted on page 6.

Notably, the second reviewer recommended removing the number of additional definitions; also, considering that we have used Pythagorean fuzzy definitions in this paper (not intuitionistic fuzzy); we only provided the Pythagorean fuzzy definitions.

Comment 4. Also, authors should add definitions of Pythagorean fuzzy set and properties  instead of add explain in Section 2.

Respond 4. Thanks for your valuable mention. The formulas were checked and revised. As an example, equation 13 was rewritten in fuzzy form to further assist the paper readers. These modifications were highlighted on page 11. The presentation structure of preliminaries was changed in the form of "definition", and these modifications were highlighted on pages 7 and 8.

Because the second respected reviewer emphasized the removal of additional definitions and recommended the presentation of the definitions used in the research, we have done this.

Comment 5. Comparison with recent study and methods (especially with PFS-MEREC methods and PFS-MACROS instead of TOPSIS method) would be appreciated. What is the motivation of the proposed paper. Also, Sensitivity analysis and discussion of results are very short.

Respond 5. Thanks for your comment. In this section, we used the MEREC to calculate the criteria weights and the MARCOS to rank the alternatives. For this reason, it was impossible to compare these findings one by one with the proposed combined methods that consider both weight calculation and ranking. Also, the proposed method was compared with PF-Entropy-TOPSIS and PF-Entropy-VIKOR; both weight calculation and alternative ranking were considered in these methods. These modifications were presented on page 4. Also, the results were shown in Table 18 on page 28 and Figure 14 on page 29.

Comment 6. Authors should add especially more PFS-Merec methods and PFS-MACROS in Table 1.

Respond 6. Thanks for your comment. This was done and highlighted in Table 1. This change was highlighted on page 5.

Comment 7. Authors should add these references for fuzzy set theory, PFS theory and applications:

- Gülşen, K. U. M., SÖNMEZ, M. E., & KARGIN, A. (2022). An Alternative Process for Determining Erosion Risk: The Fuzzy Method. Coğrafya Dergisi, (44), 219-229.

- Mishra, A. R., Rani, P., Pamucar, D., & Saha, A. (2023). An integrated Pythagorean fuzzy fairly operator-based MARCOS method for solving the sustainable circular supplier selection problem. Annals of Operations Research, 1-42.

- Ali, J. (2022). A q-rung orthopair fuzzy MARCOS method using novel score function and its application to solid waste management. Applied Intelligence52(8), 8770-8792..

- Fan, J., Wang, M., & Wu, M. An extended MEREC-EDAS approach with linguistic pythagorean fuzzy set for selecting virtual team members. Journal of Intelligent & Fuzzy Systems, (Preprint), 1-21.

Respond 7. Thanks for your mention. These citations and modifications have been applied. They were highlighted on pages 3, 4 and 5.

Reviewer 7 Report

Comments and Suggestions for Authors

Please read the attachment. Thank you.

Comments on the Quality of English Language

Please read the attachment. Thank you.

Author Response

Reviewer #7:

Please read the attachment. Thank you.

General Comments:

The manuscript presents an interesting and potentially valuable contribution to

the field of decision support systems and project selection in organizations. The

use of Pythagorean fuzzy sets and multiple-criteria decision-making methods to

address uncertainty and various criteria in project selection is a novel approach.

However, there are some areas in which the manuscript could be improved to

enhance its clarity and rigor.

Specific Comments:

Comment 1. Title: Please shorten it to no more than 10 words.

Respond 1. Thanks for your mention. The title of the paper has been corrected. We had to add it because of the opinion of the other reviewer, who said that the title should be increased.

Comment 2. Keywords: the keywords should not repeat the words or phrases in the title.

Respond 2. Thanks for your comment. Keywords reflect our paper, so we announced the basic concepts and innovations of our paper in the form of keywords.

Comment 3. All equations should be cited in the text.

Respond 3. Thanks for your comment. Explanations of all equations were given in the preliminary section. In order to make it more clear for the readers, we have changed equation 13 on page 11 to fuzzy form. On pages 22 to 26, we have added the formula related to each part during the execution of the case study to make it more clear for the readers.

Comment 4. Introduction: please rewrite the last paragraph of the introduction section.

Respond 4. Thanks for your comment. This paragraph has been revised and was highlighted on page 6.

Comment 5. All equations should be marked with a number and aligned right.

Respond 5. Thanks for your comment. This was done, and all the equations were arranged to the right.

Comment 6. Figure 4: the second box seems to lack one line side.

Respond 6. Thanks for your comment. This was done, and its color was corrected.

Comment 7. Figures 7-10: These are the screenshots in the software. I think they should

be removed from the manuscript.

Respond 7. Thanks for your mention. This transfer was done, and the above items were moved to Appendix A. This change was shown on page 31.

Comment 8. Citation [18] seems to be unnecessary. Please remove it. Also, the citations

in the main text and the author's name are wrong, too. The reference [18]

study about the EDAS approach; however, in this mention, the author listed

the studies related to MEREC and MARCOS methods.

Respond 8. Thanks for your mention. This item was removed from the research.

Comment 9. Table 1: Please take note of what "*" represents for which mentioned things.

Respond 9. Thanks for your mention. This was done and explained in the form of highlights in the paragraph below of Table 1. This change was determined on page 5.

Comment 10. Could you please provide the raw data of the actual case studies?

Respond 10. Thanks for your mention. Yes. Complete data was available to everyone in the decision support system on the web. For this purpose, you can refer to the system address with the following account and use the data below for your kind attention.

www.wdss.ir

User:dss

Pass:123

Comment 11. Why did you choose only 3 alternatives? Could you please increase the

number of other options?

Respond 2. Thanks for your comment. Because the above was a real case and according to the suggestion of Mineral Holding, this was done to test the system in a real environment. Our decision support system has the ability to be customized for different industries, with different criteria and options, the number of multiple projects, the number of multiple decision makers; and we think that these features can add to the attractiveness of its use. In addition, its most important feature was being on the web, which makes it available to everyone at any time and place through the Internet.

Comment 12. Figures: some figures' quality is too poor. Please increase their resolution.

Respond 12. Thanks for your comment. This was done. The quality of figures that were not desirable was improved, e.g., Figures 5, 8, 9, 10 and 11.

Comment 13. All equations should be the same size as the main text.

Respond 13. Thanks for your comment. The corrections were made.

Comment 14. Literature review:

 There are many MCDM and optimization approaches such as DEA, AI, FEA with GREY, VIKOR, EDAS, ANFIS, Fuzzy logic, etc.

the following papers could help:

+ Enhancing Efficiency and Cost-Effectiveness: A Groundbreaking BiAlgorithm MCDM Approach

+ Enhancing Lithium-Ion Battery Manufacturing Efficiency: A Comparative Analysis Using DEA Malmquist and Epsilon-Based Measures

Respond 14. Thanks for your comment. The considered studies were added to our revised paper and used. They were highlighted on page 3.

Comment 15. References:

+ Please check reference [36, 44]. It seems to have not followed the guidance

of the journal template.

+ Please format following the guidelines of the journal template.

Respond 15. Thanks for your comment. The corrections were made.

Comment 16. Please provide the criteria matric of the decision matric.

Respond 16. Thanks for your mention. The whole decision matrix was given in Figures 11 and12, and Table 4. These items were specified on pages 19, 20, and 21.

Comment 17. Please add a comparison of the results of the proposed method with the other previous results.

Respond 17. Thanks for your mention. This comparison was made with two other decision methods and was shown in Table 18 and Figure 14. They were specified on pages 28 and 29.

Comment 18. Please rearrange the manuscript more coherently and logically

Respond 18. Thanks for your mention. A basic and complete review was done in the entire research, and logical corrections were made.

The abstract was strengthened. It appears on page 1. The literature review was enriched. It appears on pages 2 to 5. We added research questions as highlighted on page 6. In addition, future research will be enriched; It appears on page 1. Modifications were made to the structure of the presentation of preliminary explanations on page 7. A flowchart was added in order to clarify the step-by-step implementation of the methodology on page 10. In order to further help the readers clarify the steps of MEREC method, formula 13 on page 11 of the crisp state was rewritten to fuzzy. In the case study section, more explanations were added on page 17. Managerial implications were added on page 29, and future research was enhanced on page 30.

Constructive Questions:

Comment 19. How were the Pythagorean fuzzy sets, MEREC, and MARCOS methods

applied step by step in the development of the decision support system for project selection? Can you provide a detailed algorithm or flowchart to illustrate the process?

Respond 19. Thanks for your mention. The flowchart of the paper process was added as Figure 3. The main work of paper was shown in this flowchart; It appears on page 10.

Comment 20. What practical guidance or insights can decision-makers draw from the

research findings? How can organizations effectively implement the proposed methodology in their project selection processes to improve decision-making?

Respond 20 Thanks for your mention. The advantages and comprehensiveness of our paper and its difference from other articles were explained in more detail on pages 2 and 5. In addition, managerial implications were added in our revised paper in section 7 on page 29. Our decision support system has the ability to be customized for different industries, with different criteria and options, the number of multiple projects, and the number of multiple decision makers; we think that these features can add to the attractiveness of its use. In addition, its most important feature was being on the web, which makes it available to everyone at any time and place through the Internet.

Round 2

Reviewer 3 Report

Comments and Suggestions for Authors

The authors answered all my comments.

Comments on the Quality of English Language

Minor English check is required.

Author Response

Reviewer #3:

Comments and Suggestions for Authors

The authors answered all my comments.

Comment 1. Comments on the Quality of English Language

Minor English check is required.

Respond 1. Thanks for your comment. The second revised manuscript was reviewed by an English expert, and many changes were made.

Reviewer 4 Report

Comments and Suggestions for Authors

The authors have rvised paper based on all my comments. The answers to my comments are provided as well. I do not have more comments and think that paper can be published in current form.

Author Response

Reviewer #4:

The authors have revised paper based on all my comments. The answers to my comments are provided as well. I do not have more comments and think that paper can be published in current form.

Respond. Thank you for your time. Also for your valuable comments.

Reviewer 7 Report

Comments and Suggestions for Authors

Dear Editor and Authors:

 Thank you for providing the point-to-point response.

The authors have carefully and patiently corrected and answered the comments and questions. The manuscript looks perfect now and is acceptable for publication in this journal.

Minor changes: the citation [23] should not removed from the references.

Please feel free to contact me if you have further requests or concerns.

Thank you for reading.

Author Response

Reviewer #7:

Dear Editor and Authors:

Thank you for providing the point-to-point response.

The authors have carefully and patiently corrected and answered the comments and questions. The manuscript looks perfect now and is acceptable for publication in this journal.

Minor changes: the citation [23] should not removed from the references.

Please feel free to contact me if you have further requests or concerns.

 Thank you for reading.

Respond. Thanks for your mention. During our review, reference 23 was not removed from the primary version of this paper. Rather, due to the increase and changes in references, reference 23 has been changed to reference 28 in the new numbering. This reference is highlighted in the text and references section.